# Genetic regulation of *TERT* splicing affects cancer risk by altering cellular longevity and replicative potential

Oscar Florez-Vargas [1], Michelle Ho [1], Maxwell H. Hogshead [1], Brenen W. Papenberg [1], Chia-Han Lee[1], Kaitlin Forsythe [1], Kristine Jones [2], Wen Luo[2], Kedest Teshome[2], Cornelis Blauwendraat [3], Kimberley J. Billingsley[3], Mikhail Kolmogorov [4], Melissa Meredith [5], Benedict Paten [5], Raj Chari [6], Chi Zhang[2], John S. Schneekloth[7], Mitchell J. Machiela [8], Stephen J. Chanock [9], Shahinaz M. Gadalla [10], Sharon A. Savage [10], Sam M. Mbulaiteye [11] & Ludmila Prokunina-Olsson [1]✉

The chromosome 5p15.33 region, which encodes telomerase reverse transcriptase (TERT), harbors multiple germline variants identified by genome-wide association studies (GWAS) as risk for some cancers but protective for others. Here, we characterize a variable number tandem repeat within *TERT* intron 6, VNTR6-1 (38-bp repeat unit), and detect a strong link between VNTR6-1 alleles (Short: 24-27 repeats, Long: 40.5-66.5 repeats) and GWAS signals rs2242652 and rs10069690 within *TERT* intron 4. Bioinformatics analyses reveal that rs10069690-T allele increases intron 4 retention while VNTR6-1-Long allele expands a polymorphic G-quadruplex (G4, 35-113 copies) within intron 6, with both variants contributing to variable *TERT* expression through alternative splicing and nonsense-mediated decay. In two cell lines, CRISPR/Cas9 deletion of VNTR6-1 increases the ratio of *TERT*-full-length (FL) to the alternative *TERT-β* isoform, promoting apoptosis and reducing cell proliferation. In contrast, treatment with G4-stabilizing ligands shifts splicing from *TERT-FL* to *TERT-β* isoform, implicating VNTR6-1 as a splicing switch. We associate the functional variants VNTR6-1, rs10069690, and their haplotypes with multi-cancer risk and age-related telomere shortening. By regulating *TERT* splicing, these variants may contribute to fine-tuning cellular longevity and replicative potential in the context of stress due to tissue-specific endogenous and exogenous exposures, thereby influencing the cancer risk conferred by this locus.

At least ten independent GWAS signals within the ~100 kb genomic region on chromosome 5p15.33 harboring *TERT* and *CLPTM1L* have been associated with cancer risk or protection[1–4]. *TERT* encodes the catalytic subunit of telomerase, a reverse transcriptase that extends telomeric repeats at chromosome ends to maintain telomere length and genome integrity[5], with telomerase dysfunction implicated in many human diseases[6]. *CLPTM1L* encodes a putative oncogene that promotes cancer cell growth and resistance to apoptosis[7,8]. GWAS-identified signals might be causal or tag some known or yet unknown functional polymorphisms. Thus, identifying these variants and the

mechanisms underlying their associations may improve the understanding of the etiology and biology of these cancers, leading to optimized cancer risk prediction, prevention, and treatment.

Several variable number tandem repeats (VNTRs) have been reported within the 5p15.33 region[9,10], but their characterization has been limited due to high variability, complexity, and length of genomic fragments extended by repeats. Advances in long-read genome sequencing and assembly[11] have resolved many genomic gaps, enabling deeper exploration of complex regions such as VNTRs, which remain challenging to analyze with short-read whole-genome sequencing (WGS) or PCR-based methods. Recent examples[12] have shown that VNTRs might account for or contribute to GWAS signals for cancer and other human traits, expanding the list of potentially functional variants to consider.

Here, hypothesizing that VNTRs might be responsible for some of the 5p15.33 GWAS signals, we explore two VNTRs within *TERT* intron 6 in relation to multi-cancer GWAS signals reported in this region. Among these signals, we detect a strong link only between VNTR6-1 and two single nucleotide polymorphisms (SNPs)−rs2242652 and rs10069690−within *TERT* intron 4. Specifically, we preferentially link VNTR6-1 Long alleles (40.5–66.5 repeats), in contrast with Short alleles

(24–27 repeats), with the rs2242652-A and rs10069690-T alleles, both of which were previously associated with a reduced risk of bladder[4] and prostate cancer[13] but an elevated risk of glioma[14], breast[15,16], and ovarian cancer[17]. We present a comprehensive genetic and functional analysis of VNTR6-1 and its linked GWAS signals.

## Results

### VNTR6-1 is linked with multi-cancer GWAS signals rs2242652 and rs10069690

We explored two previously reported but minimally characterized VNTRs[9,10] within *TERT* intron 6 in relation to all cancer-related GWAS signals within the 5p15.33 multi-cancer region[1-4]. First, we analyzed 452 long-read WGS assemblies from 226 controls of diverse ancestries generated by the Human Pangenome Reference Consortium (HPRC)[11] and the Center for Alzheimer's and Related Dementias (CARD)[18]. The strongest associations were detected for VNTR6-1 (38-bp repeat unit, range 24–66.5 repeats in the assemblies), with more repeats detected in assemblies with the rs2242652-A ($p = 5.93E\text{-}19$) and rs10069690-T ($p = 5.40E\text{-}11$) alleles compared with the alternative alleles at these SNPs located within the *TERT* intron 4 (Fig. 1a, b, Supplementary Fig. 1 and Supplementary Data 1). In contrast, VNTR6-2 (36-bp repeat unit, range

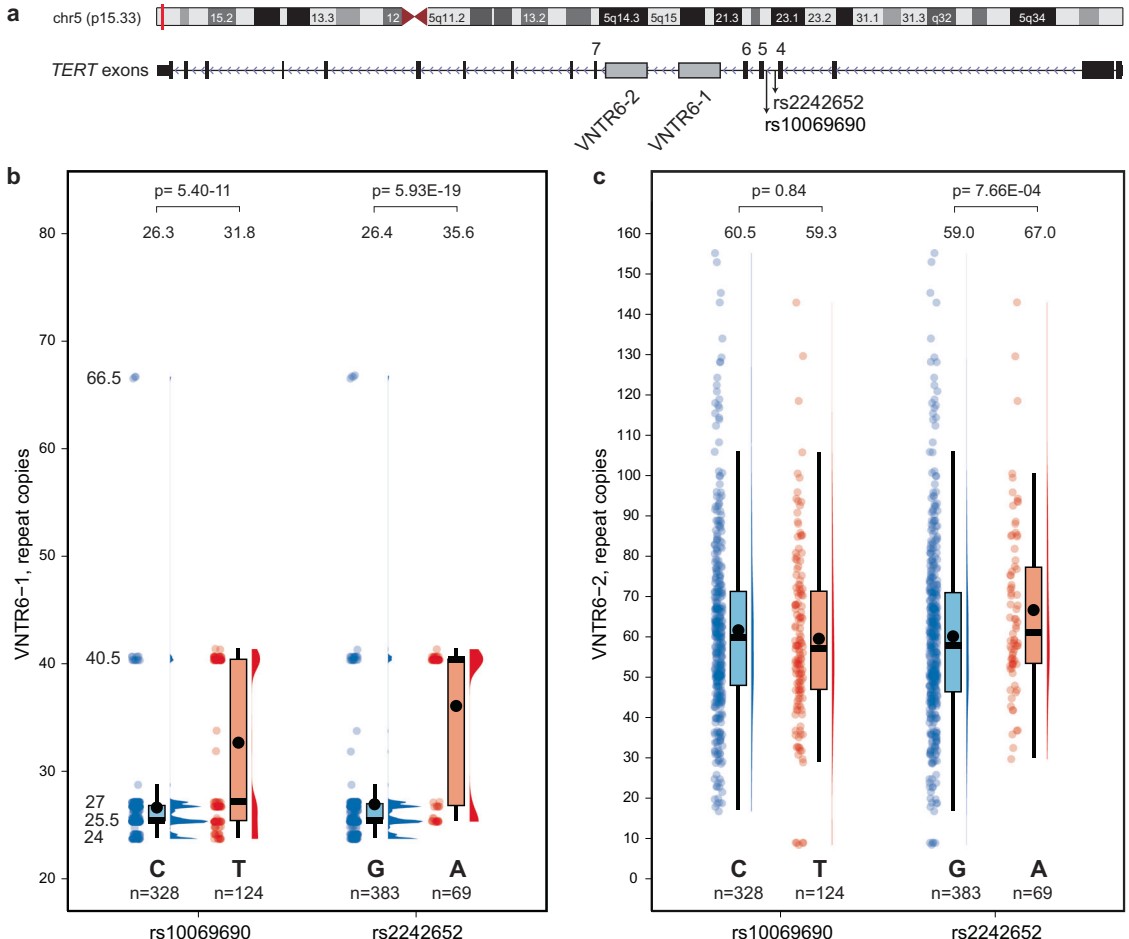

**Fig. 1 | Analysis of VNTR6-1 and VNTR6-2 within *TERT* intron 6 in relation to the multi-cancer GWAS signals rs10069690 and rs2242652. a** The chr 5p15.33 genomic region with GWAS signals rs10069690 and rs2242652 within *TERT* intron 4 and VNTRs within intron 6. **b, c** Distribution of repeat copies in 226 controls of diverse ancestries (*n* = 452 long-read WGS assemblies, with individual sample sizes for each group indicated in the figure) for (**b**) VNTR6-1 (38-bp repeat unit) and (**c**) VNTR6-2 (36-bp repeat unit). The dots represent repeat copies for each chromosome assembly. The box plots define the minima and maxima (ends of the whiskers), the center (median, shown as a horizontal black line), the bounds of the box

(first and third quartiles, representing the interquartile range), and the means (black dots, with values displayed above the corresponding plots). Half-violin plots show the density distribution of the data. Five VNTR6-1 alleles −24, 25.5, 27, and 40.5 repeats −were observed above the 5% frequency threshold and accounted for 90.04% of all alleles in the set; VNTR6-2 alleles were scattered between 8 and 155 repeat copies, all under the 5% threshold. *P*-values were calculated for unpaired two-sided Wilcoxon–Mann–Whitney tests comparing the number of repeat copies between the genotype groups. The source data are provided in the Source Data file.

8-155 repeats in the assemblies) was moderately associated with rs2242652-A ($p = 7.66E-04$) and some other GWAS signals, but not rs10069690-T allele ($p = 0.84$, Fig. 1c and Supplementary Data 1). Thus, we focused on VNTR6-1 as a potential proxy for the multi-cancer GWAS signals rs2242652 and rs10069690.

Since long-read WGS resources are limited, we performed long-read targeted sequencing of the VNTR6-1 PCR amplicon (2126-3750 bp) in various samples (Supplementary Data 2). This analysis confirmed the concordance in repeat scoring between targeted sequencing and WGS for ten HPRC controls with available assemblies[11], as well as between five bladder tumors and paired tumor-adjacent normal bladder tissues. In addition, it confirmed Mendelian segregation of VNTR6-1 alleles in HapMap samples from 30 European (Central European from Utah, CEU) and 30 African (Yoruba, YRI) family trios. Although reliable, long-read sequencing is an expensive, labor-intensive, and low-throughput method that requires significant amounts of high-quality DNA. To facilitate its evaluation in large-scale association studies, we explored additional approaches for VNTR6-1 analysis.

We noted that in both the assemblies and HapMap samples with targeted sequencing data, VNTR6-1 alleles clustered into two main groups (Fig. 1, Supplementary Data 2). In the HapMap samples, these groups included alleles designated as Short (CEU: mean $25.8 \pm 1.0$ repeats, 83.3%; YRI: mean $27.0 \pm 2.0$ repeats, 80%) and Long (CEU: mean $40.5 \pm 0$ repeats, 16.7%; YRI: mean $43.75 \pm 8.8$ repeats, 20%), with an uncommon 66.5-repeat allele detected at 2.5% frequency only in African-ancestry individuals (Supplementary Data 2). We also explored short-read WGS profiles for all individuals of diverse ancestries from the 1000 Genomes Project (1000 G, $n = 3201$).

WGS profiles of samples with only the VNTR6-1-Short alleles (24–27 repeats based on the assemblies or targeted sequencing) appeared similar to the reference human genome (GRCh38, Short allele with 27 repeats), whereas prominent read pileups were observed in samples with at least one copy of the Long allele (40.5-66.5 repeats), with no further separation within these groups (Supplementary Fig. 2a). A supervised-learning approach classified all the 1000 G samples as carriers of at least one copy of the VNTR6-1-Long allele (Long/any genotype) or the Short/Short genotype (Supplementary Fig. 2b, c and Supplementary Data 3). Treating these classifications as true VNTR6-1 genotypes, we used all SNPs within the 400 kb genomic region (GRCh38 chr5:1,100,000-1,500,000) to construct a random forest classifier across all 1000 G samples. This analysis identified two SNPs, rs56345976 and rs33961405, located 704 bp apart within the VNTR6-1 amplicon, as the best predictors of VNTR6-1 groups across populations despite differences in linkage disequilibrium (LD) profiles (Supplementary Fig. 3, Supplementary Data 4 and Supplementary Data 5). Although these SNPs were not sufficiently informative individually, the rs56345976-A/rs33961405-G haplotype separated the carriers of VNTR6-1-Long alleles from those with the VNTR6-1-Short/Short genotypes defined by three other haplotypes of these two SNPs (AA, GG, and GA) (Supplementary Fig. 4 and Supplementary Data 3). This classification was consistent with the repeat sizes determined by the long-read assemblies and targeted sequencing (Supplementary Fig. 5 and Supplementary Data 6).

We also created a custom imputation reference panel of the 400-kb region in all 1000 G samples. VNTR6-1 was incorporated into this panel as a biallelic marker with Short/Long alleles determined based on rs56345976/rs33961405 haplotypes (Supplementary Data 3). The 1000 G dataset was randomly split into two equal groups, using the first group as a reference for imputation in the second group, achieving 99.3% concordance with predetermined VNTR6-1 genotypes. These results demonstrated that in WGS datasets, VNTR6-1 could be confidently imputed as a biallelic marker with Short/Long alleles. In 1000 G European populations (1000G-EUR), the VNTR6-1-Long allele was most strongly linked with rs2242652-A ($r^2 = 0.62$) and rs10069690-T ($r^2 = 0.48$, Supplementary Data 5 and Supplementary

Fig. 6), suggesting that VNTR6-1 might contribute to associations detected for these GWAS signals.

## VNTR6-1 creates an expandable G-quadruplex that modulates *TERT* splicing

VNTR6-1 is located ~3.5 kb upstream of *TERT* exon 7 (Fig. 2a). Simultaneous inclusion or skipping of exons 7 and 8 defines the *TERT*-full-length (*TERT-FL)* or *TERT-β* isoform, respectively[19]. To assess its role in this splicing pattern, we deleted the VNTR6-1 region (2241 bp in the reference human genome) by CRISPR/Cas9 editing. Partial deletion of this highly repetitive genomic region was not possible. We established three stable isogenic VNTR6-1 knockout clones (V6.1-KO) in UMUC3, a bladder cancer cell line with high *TERT* expression (DepMap transcripts per million (TPM) = 6.78; Fig. 2a and Supplementary Fig. 7a, b) and two clones in A549, a lung cancer cell line with moderate *TERT* expression (TPM = 3.63, Supplementary Fig. 8a). We considered these knockouts and their parental wild-type (WT) cell lines as VNTR6-1 extremes (none vs. ≥24 repeats), which can be used as isogenic models for VNTR6-1-Short and -Long alleles, respectively. VNTR6-1 knockout increased the inclusion of exons 7 and 8 in both cell lines, increasing the *TERT-FL* fraction from ~45% to 71% in UMUC3 (Fig. 2b, e, Supplementary Fig. 7c–f) and from 39% to 58% in A549 (Supplementary Fig. 8b, c). These results suggest that VNTR6-1 acts as a splicing switch between the *TERT-FL* (greater fraction in knockout cells) and *TERT-β* (greater fraction in WT cells).

In search of functional features that could explain our observations, we found no evidence of differential DNA methylation (Supplementary Fig. 9) or long-range chromatin interactions (Supplementary Fig. 10) involving the VNTR6-1 region. However, we noted a high G content within the 38-bp consensus repeat sequence of VNTR6-1: (5′-GGTGGGGATCTGTGGGATTGGTTTTCATGTGTGGGGTA-3′). Based on G4Hunter analysis and G4-ChiP-seq, we predicted that VNTR6-1 adopts a G-quadruplex (G4) structure in the *TERT*-sense orientation, creating 35-113 G4 copies per allele with conserved core G-containing motifs (Fig. 2a and Supplementary Fig. 11a, b).

Since a single invariable G4 upstream of VNTR6-1 has been implicated in *TERT-β* splicing[20], we hypothesized that variation in G4 copies, created by VNTR6-1-Short vs. Long alleles, may explain the observed differences in the *TERT-FL:TERT-β* ratio. To assess the role of VNTR6-1-G4 in splicing, we treated our WT cell lines, UMUC3 (Fig. 2 and Supplementary Fig. 11c–f) and A549 (Supplementary Fig. 8), as well as their respective knockouts, with two G4-stabilizing ligands. We quantified the expression of exons 6-9 (*TERT-β*) and 7-8 (*TERT-FL*) and total *TERT* in cDNA from treated and untreated cells. Treatment with the G4 ligands Pidnarulex (CX-5461)[21] or PhenDC3[22] decreased the *TERT-FL* while increasing the *TERT-β* fraction in both the UMUC3 and A549 cell lines, likely by stabilizing VNTR6-1-G4 (Fig. 2f, g and Supplementary Fig. 8d–k). These results support the role of VNTR6-1-G4 in modulating the *TERT-FL:TERT-β* ratio. A splicing isoform with exon 8 skipping (*TERT-Δ8*, Supplementary Fig. 12) was observed in knockout and WT cells after G4 ligand treatment.

## rs10069690-T and VNTR6-1-Long alleles affect *TERT* expression and splicing

*TERT* expression is generally lower in normal human tissues (The Genotype-Tissue Expression (GTEx) Project, median TPM = 0.00–2.73) and is not associated with the GWAS signals rs2242652 and rs10069690 (Supplementary Data 7). However, *TERT* expression is generally higher in tumors (The Cancer Genome Atlas (TCGA), median TPM = 0.02–5.71; Supplementary Data 7) and is associated with these SNPs in some tumor types (kidney chromophobe, KICH, and head and neck squamous carcinoma, HNSC; Supplementary Data 7). Notably, we detected high *TERT* expression (mean TPM = 59.7, Fig. 3a and Supplementary Data 8) in a set of 78 Burkitt lymphoma (BL) tumors[23]. BL is an aggressive pediatric cancer originating from germinal center B cells,

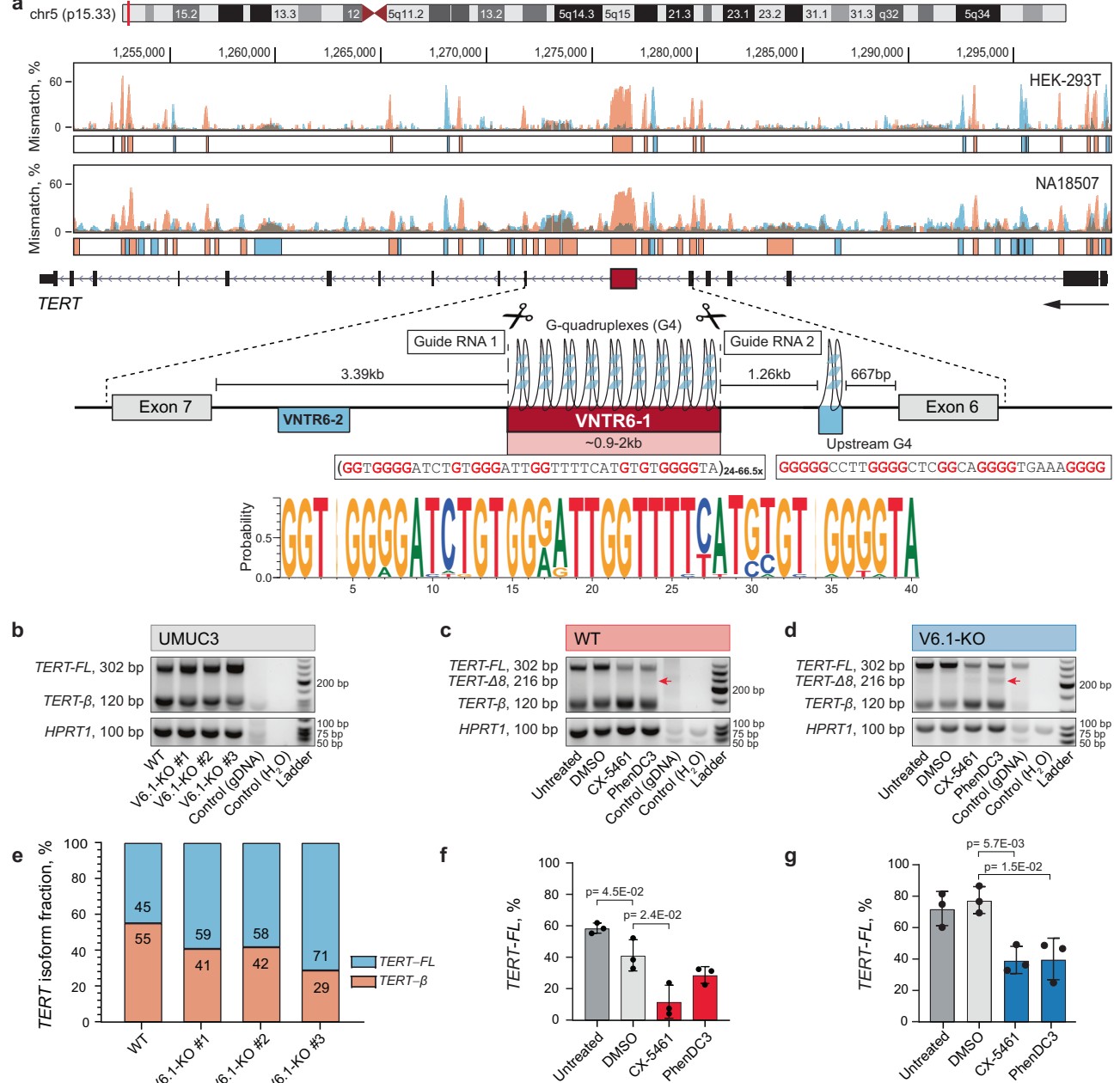

**Fig. 2 | VNTR6-1 affects the *TERT-FL*:*TERT-β* splicing ratio. a** G4-ChIP results within the *TERT* region in the cell lines HEK-293T (VNTR6-1: Long/Long) and NA18507 (YRI, 1000 G, VNTR6-1: Short/Short) display mismatches (%) during DNA synthesis, reflecting polymerase stalling after G stabilization in both the plus (blue) and minus (orange, direction of *TERT* transcription) genome strands. The genomic region of *TERT* intron 6 includes VNTR6-1 (24-66.5 copies of the 38-bp repeat unit), VNTR6-2, G4 in the minus strand (polymorphic G4 within VNTR6-1 and constitutive upstream G4), and CRISPR/Cas9 guide RNAs for excising VNTR6-1. The sequence logo shows the consensus 38-bp VNTR6-1 repeat unit in UMUC3 cells based on long-read WGS. **b** Agarose gels of RT–PCR products amplified from the cDNA of corresponding samples; gDNA–genomic DNA negative control; *HPRT1*-endogenous normalization control. **e** Densitometry results of the PCR amplicons in the plot (**b**).

The differences in the *TERT* isoform ratios are further explored in Supplementary Fig. 11. Experiments in UMUC3 cells comparing *TERT* splicing and isoform-specific expression after 72 h of treatment with G4 stabilizing ligands, normalized to *HPRT1* in the WT (**c**, **f**) and V6.1-KO (**d**, **g**) cell lines. **c**, **d** A representative agarose gel of SYBR-Green RT–qPCR products detecting several isoforms with primers located in exons 6 and 9. The extra PCR band, marked by a red arrow in panels (**c** and **d**), is further explored in Supplementary Fig. 12. **f**, **g** Densitometry analysis of the corresponding agarose gels evaluating *TERT-FL* (%) relative to the total PCR products. All analyses are based on three independent experiments, presented as means ± SD. One representative gel per experiment is shown. Comparisons were made against the vehicle control (DMSO). *P*-values are for unpaired two-sided Student's *T* test. The source data are provided in the Source Data file.

in which high *TERT* expression is necessary for the longevity of memory B cells. Two hotspot somatic mutations in the *TERT* promoter, C228T (-124 bp) and C250T (-146 bp) upregulate *TERT* expression in many tumors[24,25], but these mutations are absent in non-Hodgkin lymphomas, including BL[26] and our set of BL tumors. The combination of high *TERT* expression in the absence of upregulating promoter

mutations in BL tumors provides an opportunity to explore the regulation of *TERT* expression by germline variants.

In BL tumors (Supplementary Data 8), *TERT* expression decreased with the rs10069690-T allele (β = − 13.95 TPM, *p* = 0.035; Fig. 3b) but not with the rs2242652-A allele (β = 2.53 TPM, *p* = 0.83; Fig. 3c), with a suggestive trend for decreased *TERT* expression for the VNTR6-1-Long

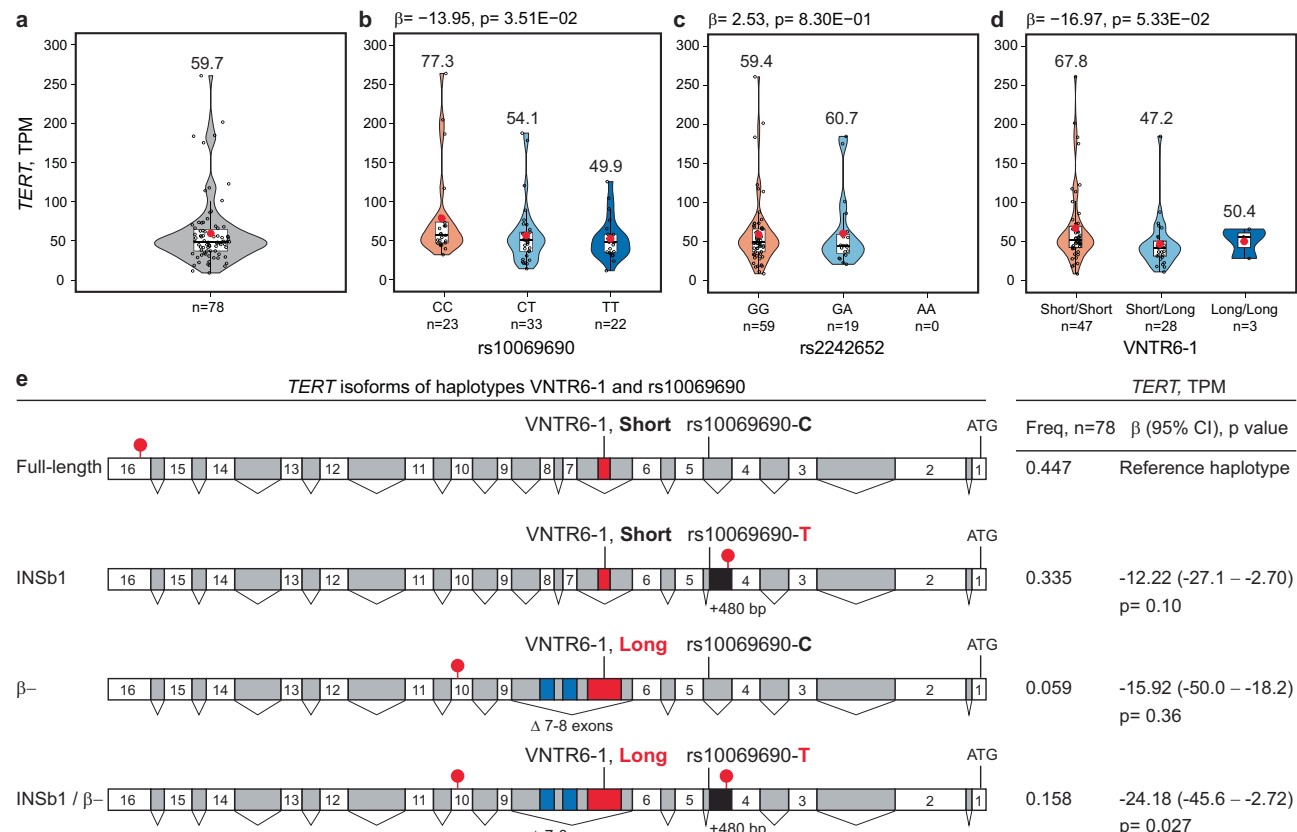

**Fig. 3 | Analysis of *TERT* expression in 78 Burkitt lymphoma (BL) tumors.** Total *TERT* expression analyzed as transcripts per million (TPM) in BL tumors (*n* = 78) (**a**) overall and in relation to the (**b**) rs10069690, (**c**) rs2242652, and (**d**) VNTR6-1 genotypes, with individual sample sizes for each group indicated in the figure. The group means are shown as red dots with values above the corresponding violin plots. Within each violin plot, the embedded box plots define the center line as the median, the whiskers as the minima and maxima, and the bounds as the 1st (25%) and 3rd (75%) quartiles of the distribution. **e** association of the VNTR6-1 and rs10069690 haplotypes with total *TERT* expression. The reference Short-C haplotype corresponds to the telomerase-encoding *TERT-FL* isoform, whereas the *INS1*

and *TERT-β* isoforms encode truncated proteins without telomerase activity. Effect alleles in haplotypes are marked in red; white boxes – exons; gray boxes – introns; black boxes – intron 4 retention; blue boxes – alternative exons 7 and 8; and red lollipops – stop codons. The direction of the *TERT* exons is from right to left, corresponding to the minus strand, as presented in the UCSC browser. "ATG" indicates translation start codons. *P*-values and β-values are for linear regression models adjusted for sex and age, which were assessed for robustness through permutation testing, with results provided in Supplementary Data 8. The details are provided in the Source Data file.

allele (β = − 16.97 TPM, *p* = 0.053; Fig. 3d). These variants are in high LD in 1000G-EUR but in low LD in 1000G-AFR and our set of BL tumors (88% from African patients, Supplementary Fig. 13), suggesting independent effects of rs10069690 and VNTR6-1 on *TERT* expression. Based on the LD profiles and association with *TERT* expression in BL tumors, we functionally prioritized rs10069690 and VNTR6-1 for further analyses.

The functional role of the rs10069690-T allele has been attributed to the creation of an alternative splicing site in *TERT* intron 4, resulting in the coproduction of telomerase-functional TERT-FL and a truncated telomerase-nonfunctional INS1b isoform[27]. However, owing to low *TERT* expression in most human tissues, this relationship has not been explored in relation to 5p15.33 genetic variants[27]. In BL tumors, 26.2% of all RNA-seq reads between exons 4 and 5 were retained within intron 4, in contrast with neighboring introns 3 and 5 (with 10.5% and 8.1% of the retained reads, respectively, Supplementary Fig. 14a). The rate of *TERT* intron 4 retention was stronger associated with rs10069690 (*p* = 5.36E-09, Supplementary Fig. 14b) than with rs2242652 (*p* = 5.0E-03, Supplementary Fig. 14c). We analyzed four splicing events between exons 4 and 5, one with canonical intron 4 splicing and three involving intron 4 retention (isoforms *INS1*[19,27], *INS1b*[27], and unspliced intron 4, Supplementary Fig. 15a–d). Canonical splicing decreased across rs10069690 genotypes (68.3%, 63.8%, and 57.3% of reads in CC, CT and TT groups, respectively; *p* = 1.65E-05; Supplementary Fig. 15e). With

each copy of the rs10069690-T allele, *INS1b* splicing increased from 0% to 3.8% and 7.0% (*p* = 3.07E-09; Supplementary Fig. 15g), while *INS1* splicing decreased and unspliced intron 4 (excluding reads for *INS1* and *INS1b* isoforms) increased (Supplementary Fig. 15f, h). These results are consistent with the previously reported association between the rs10069690-T allele and *INS1b*-type splicing[27] but suggest that *INS1*- and *INS1b*-type splicing are minor and likely secondary to intron retention, which increases with the rs10069690-T allele. A similar trend was observed for rs2242652 but with weaker associations (Supplementary Fig. 15i–l).

Several other common *TERT* isoforms have been reported[19] (Supplementary Fig. 16). The *TERT-α* isoform involves in-frame 36 bp skipping within exon 6 ($\Delta6_{(1-36)}$), causing partial loss of the reverse transcriptase domain[19]. As discussed above, *TERT-β* ($\Delta7-8$)[19] results from the simultaneous skipping of exons 7 and 8 (182 bp), terminating the frameshifted protein in exon 10. In addition, *TERT-αβ* results from concurrent $\Delta6_{(1-36)}$ and $\Delta7-8$ splicing events. The expression of these *TERT* isoforms was not significantly associated with rs10069690, rs2242652, or VNTR6-1 in BL tumors (Supplementary Data 8). Transcripts truncated by premature termination codons (Supplementary Fig. 16), including *INS* (truncated within exon 5), *INS1b* (intron 4), and *TERT-β* or *TERT-αβ* (exon 10), are likely to be eliminated by nonsense-mediated decay (NMD), reducing total *TERT* expression. Escaping NMD would result in alternative TERT proteins without telomerase

activity but still binding the telomerase RNA component (TERC), thus producing dominant-negative competitors of the telomerase-functional TERT-FL[28].

Due to premature termination codons (in intron 4 for *INS1b* and in exon 10 for *TERT-β*), both rs10069690 and VNTR6-1 increase the fraction of NMD-targeted transcripts encoding telomerase-nonfunctional proteins, decreasing total *TERT* expression and the fraction of the telomerase-encoding *TERT-FL* isoform. To assess the combined effects of these variants, we analyzed *TERT* expression based on the VNTR6-1/rs10069690 haplotypes (Fig. 3e). Compared to the Short-C haplotype, *TERT* expression was decreased with the Short-T ($\beta = -12.2$ TPM, $p = 0.10$) and Long-C ($\beta = -15.92$ TPM, $p = 0.36$) haplotypes, with a greater decrease occurring when both the VNTR6-1-Long and rs10069690-T alleles were included in the same haplotype (Long-T, $\beta = -24.18$ TPM, $p = 0.027$, Fig. 3e and Supplementary Data 8). Thus, two splicing events independently contributed by the VNTR6-1-Long and rs10069690-T alleles (a splicing switch from the *TERT-FL* to *TERT-β* isoform and intron 4 retention) decrease total *TERT* expression, with stronger effects expected in the presence of both alleles.

## VNTR6-1 regulates proliferation and apoptosis

We hypothesized that variation in the *TERT-FL:TERT-β* ratio due to VNTR6-1 length could affect cellular dynamics, such as proliferation. To address this, we monitored the counts of UMUC3 WT and V6.1-KO cells over ten days using the Lionheart automated microscope. The differences in cell counts were not significant when WT and knockout cells were continuously cultured in a medium supplemented with fetal bovine serum (full medium, Supplementary Fig. 17a). However, when the cells were first cultured in a medium without any serum (serum-starved) for 24 h and then switched to a full medium, a strong increase in proliferation was observed only in the WT cells (Supplementary Fig. 17b). To further explore the role of VNTR6-1 in response to culturing conditions, we assessed cell proliferation as cell index, measured with xCELLigence as a real-time increase in cell counts. The cells were first cultured for two days in a medium supplemented with charcoal-stripped serum (CS medium, depleted of hormones and growth factors), followed by culturing in fresh media (CS or full) for ten more days. The knockout clones of both UMUC3 and A549 cell lines proliferated significantly slower than WT cells (Fig. 4a, Supplementary Fig. 18a and Supplementary Data 9). Similarly to the previous results (Supplementary Fig. 17), switching to the full medium resulted in a stronger and faster increase in proliferation in WT compared to knockout cells (Fig. 4a and Supplementary Fig. 18a). The increase in proliferation in WT versus knockout was less dramatic (Fig. 4a for UMUC3) or undetectable (Supplementary Fig. 18a for A549) in cells continuously cultured in CS medium. These results support the role of VNTR6-1 in regulating proliferation, potentially by providing adaptation in response to alterations in cellular conditions and stress, such as the availability of serum growth factors and hormones in our experiments.

Because proliferation reflects the balance between cell division and apoptosis, we analyzed both processes. We stained UMUC3 cells with an intracellular dye (CFSE) and monitored the decrease in fluorescence intensity that occurs as the cells divide. This analysis showed that even though all cells divided faster in the full medium, knockout clones divided slower than WT cells regardless of the culturing medium (Fig. 4b, c and Supplementary Fig. 17c). Annexin V staining of both UMUC3 and A549 cells revealed a significant increase in the percentage of apoptotic cells in knockout compared to WT cells cultured in CS media with cisplatin, an apoptosis-inducing DNA-damaging agent[29], with a weaker effect also observed in full media (Fig. 4d, e and Supplementary Fig. 18b, c). RNA-seq analysis of UMUC3 knockout compared to WT cells also demonstrated the role of VNTR6-1 in promoting proliferation and protection from apoptosis (Fig. 4f, g, Supplementary Data 10 and Supplementary Data 11).

TERT-β is a dominant-negative competitor of TERT-FL for telomerase function[28], but the interplay of these isoforms in proliferation is unclear. We monitored proliferation measured as cell index in a bladder cancer cell line with low *TERT* expression (5637 cells, DepMap *TERT* TPM = 1.23) after transient transfection with the *TERT-FL* or *TERT-β* plasmid (Supplementary Fig. 19a, b). Compared to the GFP control, overexpression of either isoform increased proliferation, with a weaker effect for *TERT-FL* compared to *TERT-β* (Supplementary Fig. 19c and Supplementary Data 12). Co-transfection at 20:80% and 80:20% *TERT-FL:TERT-β* plasmid ratios (modeling WT and V6.1-KO cells, respectively) also increased proliferation compared to control. However, cells transfected with more *TERT-FL* (80:20% ratio) grew slower than those transfected with more *TERT-β* (20:80% ratio, Supplementary Fig. 19d and Supplementary Data 12), potentially due to reduced levels of anti-apoptotic TERT-β.

We imaged A549 cells, in which visualization is facilitated by a large cytoplasm. In cells co-transfected with an equal ratio of both *TERT* isoforms, stronger mitochondrial colocalization was observed for TERT-β than TERT-FL (Fig. 5a and Supplementary Fig. 20). These results further suggest that TERT-β plays a role in mitochondrial-localized processes, such as protection from apoptosis, particularly under stress conditions[30,31]. Collectively, these experiments independently demonstrated that an increased *TERT-FL:TERT-β* ratio due to the loss of VNTR6-1 (in V6.1-KO cells) or carriage of the VNTR6-1-Short allele, may result in a reduction in *TERT-β* expression to levels insufficient for protection from apoptosis, thus negatively affecting proliferation. These anti-apoptotic effects might manifest only in specific conditions that increase cellular stress, such as DNA damage, nutrient deficiency, or other microenvironmental challenges (Fig. 5b).

## VNTR6-1 and rs10069690 account for multi-cancer GWAS associations

Because we linked the VNTR6-1-Long allele with the GWAS signals rs2242652-A ($r^2 = 0.62$) and rs10069690-T ($r^2 = 0.48$) in the 1000G-EUR populations, we next sought to compare the cancer associations of these markers. Having validated the rs56345976/rs33961405 haplotypes as a confident predictor of VNTR6-1 Short vs. Long alleles (Supplementary Data 3, Supplementary Data 4, Supplementary Fig. 3 and Supplementary Fig. 4), we used these haplotypes to infer VNTR6-1 alleles in various sets. In the absence of WGS data, we used data based on array genotyping and imputation, although this might reduce confidence in inferring VNTR6-1 status, as it depends on the accuracy of the imputation and phasing of rs56345976 and rs33961405. Specifically, we inferred VNTR6-1 and the composite marker VNTR6-1/rs10069690 because it captured the functional effects of both variants. Using these markers, we performed association analyses in individuals of European ancestry from the Prostate, Lung, Colorectal, and Ovarian (PLCO) cohort[32] of cancer-free controls ($n = 73,085$) and individuals with 16 cancer types ($n = 29,623$). The PLCO association results for the VNTR6-1-Long allele and VNTR6-1/rs10069690 were comparable to those for the rs10069690-T and rs2242652-A alleles; these alleles were associated with a reduced risk of bladder and prostate cancer but an elevated risk of breast, endometrial, ovarian, and thyroid cancer and glioma (Fig. 6a and Supplementary Data 13). Conditional analysis for VNTR6-1 eliminated or attenuated associations for rs2242652 and rs10069690. The minor residual associations after adjustment for VNTR6-1 could reflect the limitations in inferring its status in the absence of WGS data. Compared to the reference Short-C haplotype of the combined VNTR6-1/rs10069690 marker, the strongest cancer-specific associations, both positive and negative, were observed for the Long-T haplotype (Supplementary Fig. 21a).

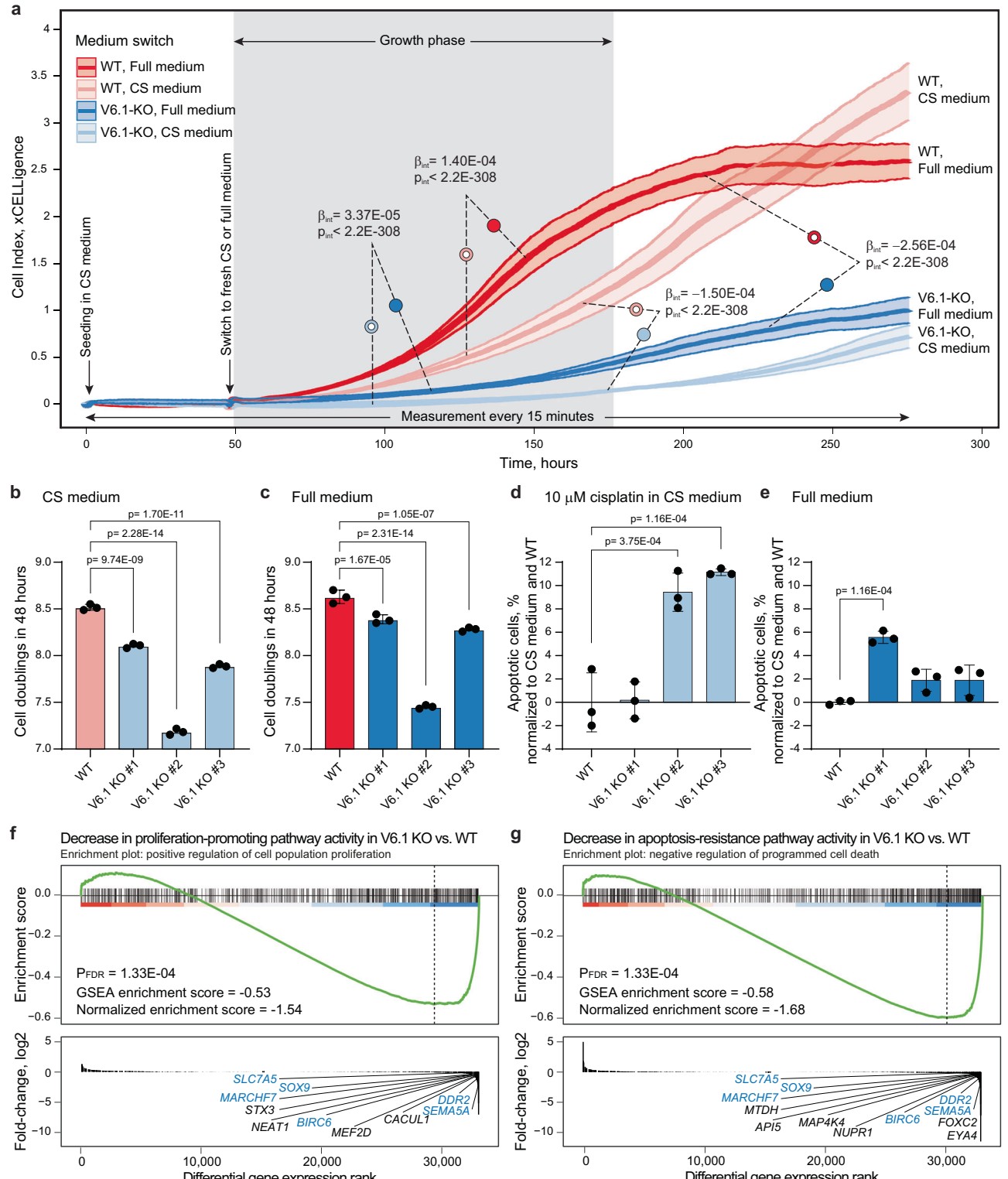

## Associations of *TERT* isoforms and genetic variants with telomerase-related metrics

*TERT-β*, which encodes a telomerase-nonfunctional protein, is the major *TERT* isoform in both normal and tumor tissues (Supplementary Fig. 22). Of the total *TERT* expression, the *TERT-FL* and *TERT-β* isoforms represented on average 17.7% and 67.9%, respectively, in 30 normal tissue types in GTEx, while 38.4% and 44.5%, respectively, in 33 tumor types in TCGA (Supplementary Data 14). To further explore the functional differences between TERT-FL and TERT-β, we assessed four

telomerase-related metrics: EXpression based Telomerase ENzymatic activity Detection (EXTEND)[33], stemness (mRNAsi)[34], the telomerase signature score[35] and telomere length in primary tumors[35] (Supplementary Fig. 22 and Supplementary Data 15). In GTEx, significant correlations with the EXTEND signature (positive for *TERT-FL* and negative for *TERT-β*) were observed in four tissues (blood, colon, esophagus, and testis). Similarly, in TCGA, most tumors with significant correlations across all metrics showed positive values for *TERT-FL* and negative values for *TERT-β*. In TCGA, of the four metrics, telomere length in

**Fig. 4 | VNTR6-1 affects proliferation and apoptosis in the bladder cancer cell line UMUC3. a** Analysis of real-time increase in cell counts (cell index) measured with xCELLigence over 283 h in UMUC3 cells. The WT and knockout cells (clone #2, starred samples in Supplementary Data 9) were cultured in a CS medium for 48 h, followed by a switch to fresh CS or full medium for 10 more days. Proliferation rates in response to culturing conditions were significantly lower in V6.1-KO compared to WT cells. The plots present the results of one of three independent experiments, with means ± SEM from $n$ = 6 biological replicates. Statistical significance and β-values for differences in the cell index during the visually determined growth phase (gray highlighting between 50 and 183 h) were calculated using linear mixed-effects interaction models. The reference sample is labeled with a dotted circle; $\beta_{int}$ represents the change in growth rates between experimental groups. **b, c** Analysis of cell doublings in UMUC3 cells cultured in (**b**) CS medium or (**c**) full medium for four days, using the CFSE assay described in Supplementary Fig. 18c. Data

presented are the results of one of three independent experiments, with means ± SD from $n$ = 3 biological replicates. $P$-values in graphs are for two-way ANOVA using Tukey's test. **d, e** Quantification of apoptosis in UMUC3 cells cultured for 48 h (**d**) with 10 μM cisplatin in CS medium or (**e**) in full medium, followed by Annexin V/PI staining and flow cytometry analysis to determine the percentage of apoptotic cells. Data presented are the results of one of three independent experiments, with means ± SD from $n$ = 3 biological replicates (normalized to values of CS media and the WT groups). $P$-values in graphs were determined by one-way ANOVA using Dunnett's test. **f, g** Gene set enrichment analysis (GSEA) for differential expression of genes involved in (**f**) pathways related to the downregulation of cellular processes for proliferation-promotion and (**g**) apoptosis-resistance, as identified by RNA-seq analysis comparing UMUC3 V6.1-KO (clone #3) to WT UMUC3 cells. Genes highlighted in blue are common to both pathways. The source data for all panels are provided in the Source Data file.

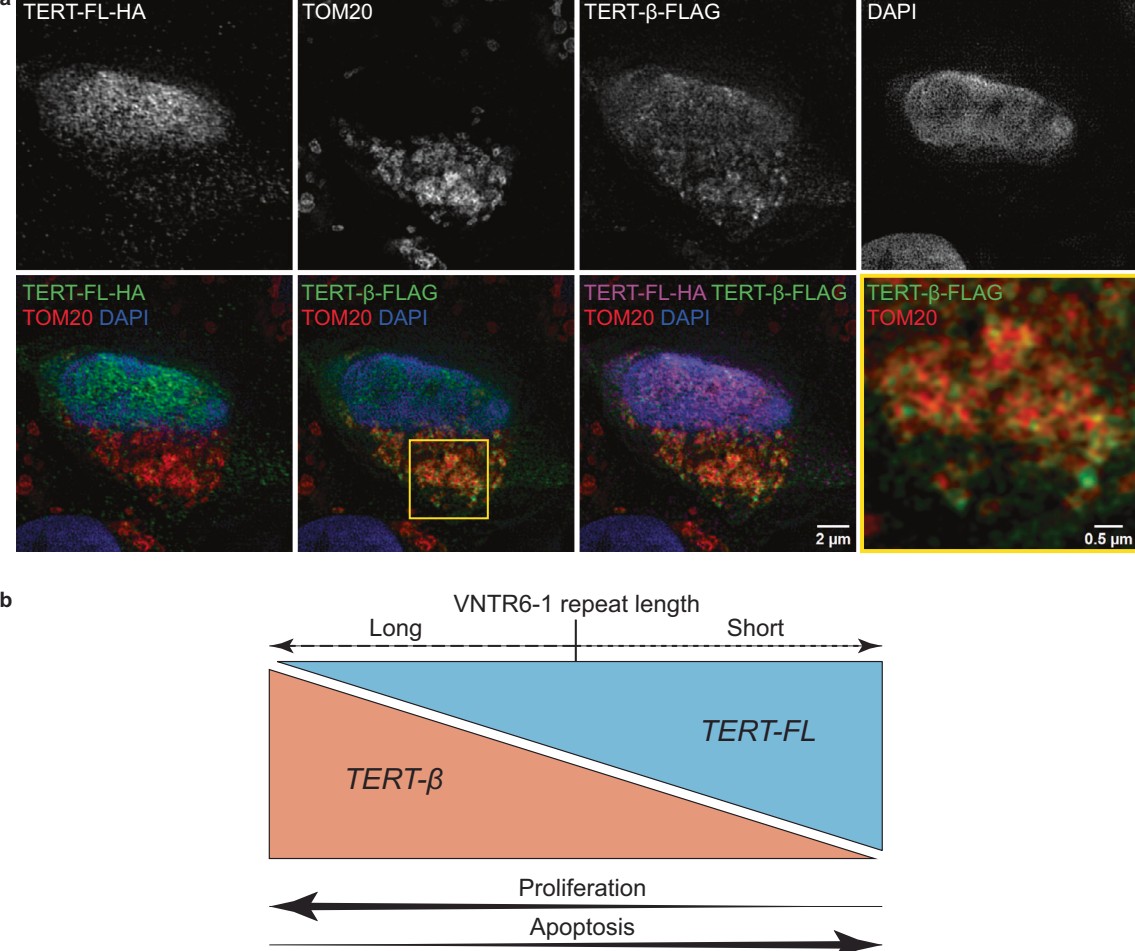

**Fig. 5 | Functional differences between the TERT-FL and TERT-β isoforms.**
**a** Structured illumination microscopy images of A549 cells co-transfected with TERT-FL and TERT-β expression constructs at a 50:50% ratio. For individual channels, the staining is shown as black/white images for better contrast. Tri-color merged panels: green – FLAG (TERT-β) or HA (TERT-FL), blue – DAPI (nuclei). Quad-color merged panel: purple - HA (TERT-FL), green - FLAG (TERT-β), red - TOM20

(mitochondria), blue - DAPI (nuclei). The yellow inset in the TERT-β-FLAG panel is shown at a higher magnification to demonstrate colocalization with mitochondria (yellow staining). The images shown are representative of ten distinct images taken across two independent experiments. Scale bar = 2 μm, inset scale bar = 0.5 μm.
**b** Schematic overview of the proposed relationships between VNTR6-1 repeat length, the *TERT-FL:TERT-β* ratio, proliferation, and apoptosis.

tumors showed the weakest correlations with *TERT* isoform expression (Supplementary Fig. 22), potentially due to somatic events, including *TERT*-upregulating promoter mutations[24,25].

Because TERT activity is essential for maintaining telomere length, we next tested the associations of VNTR6-1, rs10069690, and their haplotypes with relative leukocyte telomere length (rLTL). We inferred VNTR6-1 and VNTR6-1/rs10069690, as described above, in

cancer-free individuals of European ancestry ($n$ = 339,103) from the UK Biobank (UKB) based on SNP genotyping and imputation. In this analysis, the Short-C haplotype was associated with shorter rLTLs (β = − 0.049, $p$ = 8.75E-78, Fig. 6b, Supplementary Fig. 21b and Supplementary Data 16). Significant associations were also observed with several known markers[36,37] within *TERT* intron 2, including rs7705526 (β = − 0.079, $p$ = 1.02E-219; Supplementary Data 16) and adjustment for

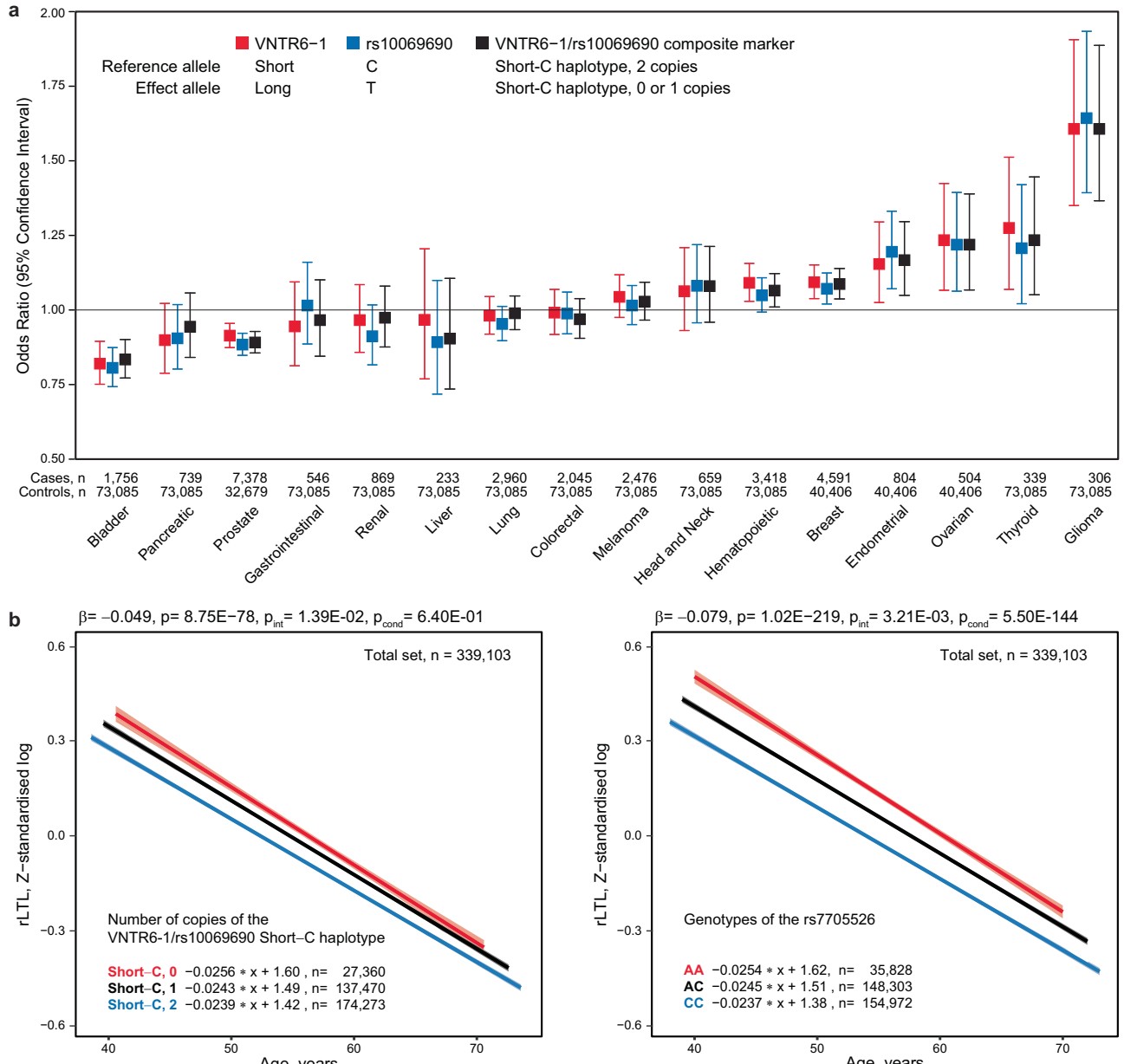

**Fig. 6 | Association analyses for cancer risk in PLCO and relative leukocyte telomere length (rLTL) in UKB cancer-free individuals. a** Evaluation of cancer risk in the PLCO dataset ($n = 102{,}708$) for the VNTR6-1-Long and rs10069690-T alleles and the composite marker (VNTR6-1/rs10069690). Odds ratios (ORs) are shown as squares with 95% confidence intervals (CIs) calculated for comparisons between patients with the indicated cancers and a common group of cancer-free controls using logistic regression analysis with an additive genetic model adjusted for sex and age. **b** Evaluation of the relationships in UKB cancer-free individuals ($n = 339{,}103$) between rLTL and VNTR6-1/rs10069690 and rs7705526 (LD, $r^2 = 0.33$ between VNTR6-1/rs10069690 and rs7705526). *P*-values and β-values were derived from linear regression models adjusted for sex, age, and smoking status. $P_{int}$ represents the interaction between genotypes and 5-year age groups; $P_{cond}$ represents the mutual adjustment for rs7705526 or VNTR6-1/rs10069690. The graphs display regression lines and equations, with a shaded area representing the 95% confidence intervals. The analysis revealed a decrease in the rLTLs with more copies of the Short-C haplotype. The results of sex-specific analyses of VNTR6-1/rs10069690 are presented in Supplementary Fig. 21b. The details are provided in the Source Data file.

rs7705526 eliminated the rLTL association with VNTR6-1/rs10069690 ($p = 0.64$). Notably, regression slopes for these markers differed by genotypes (Fig. 6b). Interaction analysis in 5-year interval groups revealed a significantly slower ($p_{int} = 1.39E{-}02$, Supplementary Data 16) decrease in the rLTL in younger individuals and a faster decrease in older individuals without the Short-C haplotype (which corresponds to a greater fraction of telomerase-nonfunctional TERT) than in carriers of this haplotype. This effect remained unchanged after adjustment for rs7705526 ($p_{int} = 1.38E{-}02$, Supplementary Data 16), which had an independent significant interaction ($p_{int} = 3.21E{-}03$, Supplementary Data 16). The rLTL association pattern was consistent in a smaller set of

healthy individuals, whose lymphocyte telomere length was measured by flow-FISH[38] and VNTR6-1 status was determined by long-read targeted sequencing, but interaction analysis was limited by sample size and age range (Supplementary Data 17).

In cancer-free controls of European ancestry, the Short-C haplotype frequencies were comparable (71.36–72.07%) across 40- to 80-year-old age groups in the UKB and PLCO but decreased to 67% in individuals aged 98–108 years (Supplementary Fig. 23). The difference between centenarians and 40- to 80-year-olds was contributed by decreased frequencies of both the rs10069690-C and the VNTR6-1-Short alleles. The VNTR6-1 genomic region is absent in non-primate

species (Supplementary Fig. 24). In genomes of several primates, as well as archaic humans (Neandertal and Denisova), only the Long-T haplotypes were observed, with the VNTR6-1 consensus repeat sequences in primates being nearly identical to those in humans (Supplementary Fig. 24 and Supplementary Fig. 25). Thus, the VNTR6-1-Short and rs10069690-C alleles, as well as the Short-C haplotype, which increase the fraction of the telomerase-functional *TERT-FL* isoform but might negatively affect longevity, are human-specific and major or common in all modern human populations (Supplementary Data 18).

## Discussion

Cancer risk is influenced by complex interactions between genetic and environmental factors. The numbers and replicative potential of stem cells in each tissue type determine the probability of acquiring mutations due to replicative errors occurring with every cell division, further modulating cancer risk[39–41]. In this work, we showed that the genetic regulation of *TERT* splicing by an SNP rs10069690 and VNTR6-1, a 38-bp intronic tandem repeat, accounts for the reduced or elevated cancer risk associated with multi-cancer GWAS leads rs2242652 and rs10069690 at 5p15.33.

While many VNTRs have been reported, including within the *TERT* region[9,10], their use in association studies remains limited. VNTRs are often highly polymorphic, with a wide range of repeat numbers that are difficult to quantify and link with biallelic markers commonly used in GWAS, such as SNPs. VNTR6-1 within *TERT* intron 6 is unusual because its repeat numbers could be binarized into two distinct allelic groups we defined as Short (24–27 repeats) and Long (40.5 or 66.5 repeats). Using multiple public and custom datasets and tools, we established VNTR6-1 as a proxy for two multi-cancer GWAS leads in this region, rs2242652, and rs10069690.

We inferred the VNTR6-1 allelic groups (Short vs. Long alleles) across diverse populations, including cancer cases and controls, based on haplotypes of the common SNPs rs56345976 and rs33961405. Although predicting VNTR6-1 status might carry more inherent technical uncertainty than the GWAS leads rs2242652 and rs10069690, our genetic analysis of cancer risk revealed comparable associations for these variants. While no functional properties were identified for rs2242652, we demonstrated that both the VNTR6-1-Long and rs10069690-T alleles are functional. Independently and in combination (i.e., Long-T haplotype), these alleles shift splicing from *TERT-FL*, which encodes telomerase, to alternative isoforms *INS1b* and *TERT-β*, which encode telomerase-nonfunctional TERT.

In addition to its canonical role in telomerase activity that is mediated by the TERT-TERC complex and supports telomere maintenance, TERT also has important non-canonical telomere-independent roles in supporting cellular homeostasis. Our findings, based on several methods and in line with previous observations[28,42], support the anti-apoptotic role of the TERT-β isoform, likely related to its mitochondrial localization and contributing to increased proliferation.

We hypothesize (Fig. 7) that the genetic regulation of the *TERT-FL:TERT-β* ratio has context-dependent consequences. The increase in the fraction of the anti-apoptotic *TERT-β* isoform extends cellular longevity (lifespan of individual cells)[43], manifesting as increased proliferation (replicative potential), especially in response to cellular stress and other stimuli. The tissues representing cancers with the most significant inverse associations for the rs10069690-T and VNTR6-1-Long alleles (such as protection from bladder cancer and risk for glioma) have low replicative potential at homeostasis. Under normal conditions, bladder epithelium is one of the slowest-growing epithelial tissues with high resistance to apoptosis[44], while the brain has limited cell-specific neurogenesis in restricted regions[45]. However, through direct contact with the urine, bladder epithelium is exposed to pathogens and reactive metabolites that can cause tissue damage and trigger acute regenerative proliferation that restores the tissue

integrity and function within days[44]. In contrast, direct exposure to damaging agents requiring tissue regeneration, as well as the capacity to regenerate, is limited for the brain. Thus, cancer susceptibility may depend not only on the replicative potential of the normal tissues at homeostasis but also on the types, timing, and intensity of damaging exposures and the ability of the tissue to regenerate after damage. The increased fraction of the TERT-β protein extending cellular longevity in bladder tissue may limit the need for tissue regeneration, thereby mitigating mutagenesis from replication errors and decreasing cancer risk. In contrast, the increased fraction of the TERT-β protein extending cellular longevity of glial cells might increase cancer risk by promoting the gradual accumulation of somatic mutations from proliferation, especially under subtle but prolonged exposures, and prevent the death of damaged cells.

In normal tissues with low replicative potential, including the bladder and brain, tumorigenesis often depends on driver mutations, such as *TERT*-upregulating promoter mutations that reactivate telomerase and immortalize cancer cells[46]. Tissue regeneration can also be initiated by rare TERT-high cells acting as stem cells[47] leading to tumorigenesis upon the acquisition of driver mutations. While the anti-apoptotic function of TERT-β is important for extending cellular longevity, the reciprocal decrease in the TERT-FL fraction might prevent immortalization of cells with acquired somatic mutations or protect against telomere shortening, especially under oxidative stress[42,48] and facilitate the DNA damage response[49].

Notably, we did not detect GWAS associations for the same alleles/haplotypes of rs10069690 and VNTR6-1 for cancers originating from tissues with high proliferation at homeostasis (e.g., the gastrointestinal tract). High proliferation rates in stem cells of these tissues, combined with cell death induced by critical telomere shortening in differentiated cells, prevent cells from reaching a malignant state and thus act as a tumor-suppressive mechanism[50]. For cancer types with no or marginal associations for the alleles/haplotypes tested, TERT-related mechanisms might be more heterogeneous and dependent on cell specificity, tumor subtype, timing, and the nature and intensity of environmental exposure.

Telomere length has been extensively studied in relation to cancer and non-cancer conditions[51,52]. Mendelian randomization analysis revealed an association between genetically predicted longer telomeres and the risk of 8 out of 22 cancer types tested, especially for rare cancers and those originating from tissues with low replicative potential[53]. Our analysis in the UKB revealed a strong association between the VNTR6-1/rs10069690 haplotypes and rLTL but weaker than those with other *TERT* rLTL markers (rs7705526, rs2736100, and rs2853677) known to be linked with telomere length[36,37]. We noted a greater degree of telomere shortening in older than in younger individuals without the Short-C haplotype. Given the anti-apoptotic role of TERT-β, which might extend cellular longevity, this could reflect a greater proportion of circulating leukocytes originating from stem cells and their progenitors that have undergone more cell divisions.

The alleles associated with an increased fraction of telomerase-encoding *TERT-FL*—VNTR6-1-Short, rs10069690-C, and their Short-C haplotype—are human-specific variants, that in Europeans are less common in centenarians than in younger individuals. The emergence and retention of these alleles might be consistent with the disposable soma theory of ageing, which postulates that evolution favors factors supporting reproductive fitness and growth at the expense of longevity, which requires substantial maintenance to repair age-related somatic damage[54]. Female fertility strongly depends on ovarian telomerase[55] and telomere shortening is considered an evolutionary cost of reproductive trade-offs[56]. The evolutionary selection of genetic variants that increase the fraction of the telomerase-encoding *TERT-FL* isoform might provide this reproductive fitness benefit while decreasing longevity later in life, perhaps due to elevated cancer risk.

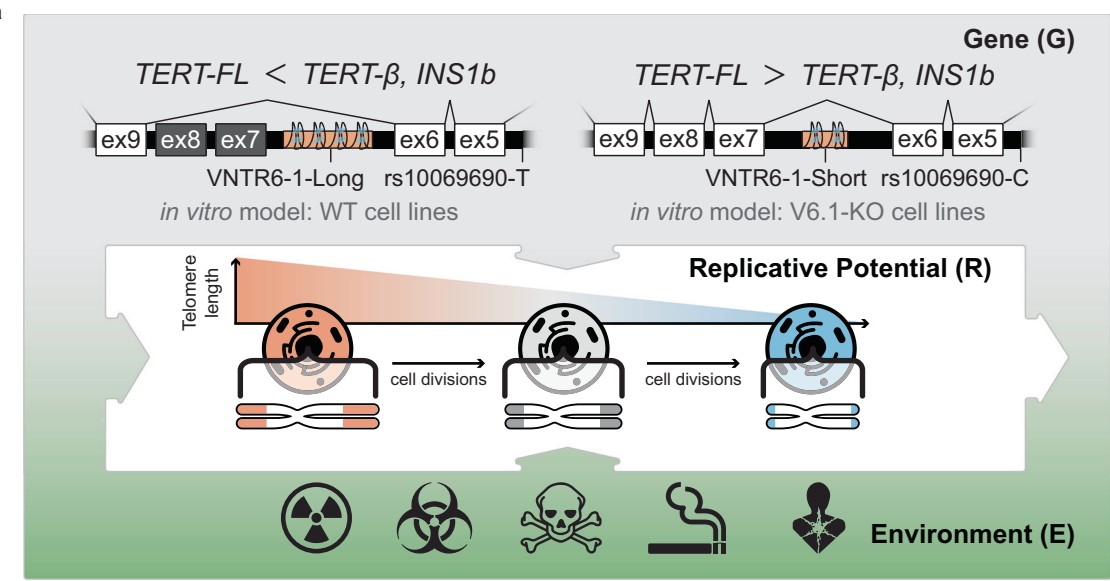

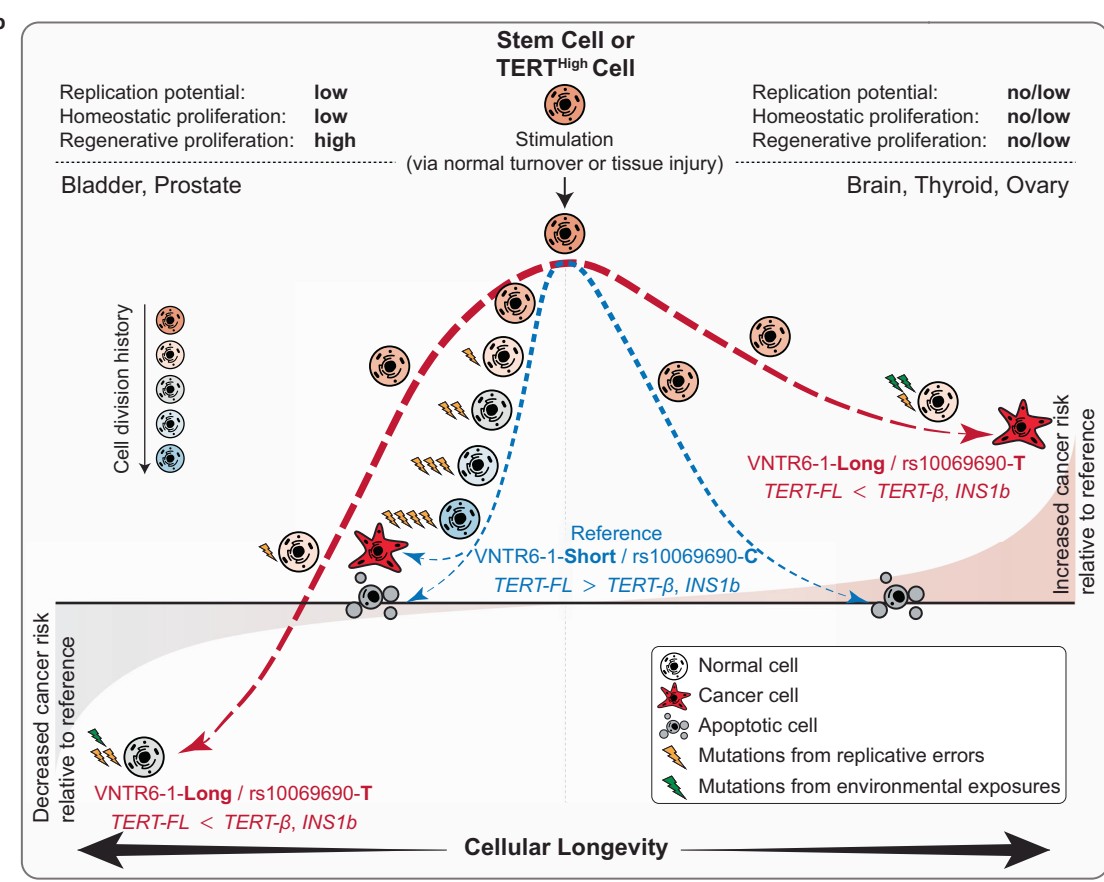

**Fig. 7 | Proposed model for functional effects of VNTR6-1 and rs10069690 contributing to multi-cancer associations within the 5p15.33 region. a** Cancer risk as a product of interactions between gene, replicative potential, and environment (G × R × E). *TERT* genetic variants VNTR6-1 and rs10069690 and environmental factors define the relative ratios of the isoforms encoding telomerase-functional *TERT-FL* and telomerase-nonfunctional *TERT-β* and *INS1b* isoforms. These isoforms affect cell proliferation, apoptosis, and telomere length, thus modulating cellular longevity and replicative potential, including homeostatic proliferation, which maintains tissue self-renewal, and regenerative proliferation, which responds to environmental factors and tissue damage. **b** The VNTR6-1-Long

and rs10069690-T alleles, or their haplotype (Long-T), are associated with reduced cancer risk in tissues with low homeostatic but high regenerative potential (e.g., bladder). The anti-apoptotic effect of the *TERT-β* isoform reduces the need for regenerative proliferation, thus decreasing the risk of acquiring mutations from replicative errors. In tissues with no/low homeostatic and regenerative proliferation (e.g., brain, thyroid, ovary), the same alleles and Long-T haplotype are associated with elevated cancer risk. The anti-apoptotic effect of *TERT-β* contributes to extended cellular longevity, allowing the accumulation of more mutations from environmental exposures, such as reactive oxygen species (ROS), cellular metabolites, etc.

Further studies are warranted to explore our findings in the context of other 5p15.33 multi-cancer GWAS signals[1,2] and identify specific splicing factors that bind to VNTR6-1. DNA:RNA hybrids, including G4 and R-loops, are emerging as important regulators under normal and disease conditions[57], and their therapeutic targeting through VNTR6-1 might be possible for modulating TERT functions. In conclusion, our multi-faceted study uncovers the complex regulation of TERT functions and multi-cancer cancer risk through a combination of *TERT* germline variants – an SNP rs10069690 within intron 4 and a VNTR within intron 6 (VNTR6-1). Our results provide insights into analyses of complex genetic variants and their contributions to cancer susceptibility and telomere biology.

## Methods

The research presented in this paper complied with all relevant ethical regulations and informed consent was obtained by each contributing study that granted access to data to perform analyses reported in this work. The study used deidentified controlled access data from the Center for Alzheimer's and Related Dementias (CARD) of the National Institute on Aging (dbGaP phs001300.v4.p1), Burkitt Lymphoma Genome Sequencing Project (BLGSP, dbGaP phs000527.v6.p2), the Prostate, Lung, Colorectal and Ovarian (PLCO) Cancer Screening Trial (project #PLCO-957), UK Biobank (project #92005) and The Cancer Genome Atlas (TCGA, https://gdc.cancer.gov). The use of deidentified bladder tissue samples was approved by the NIH Office of Human Subjects Research (#4715). The use of deidentified samples from the Center for International Blood and Marrow Transplant Research biorepository (CIBMTR; https://cibmtr.org) was approved by the National Marrow Donor Program Institutional Review Board. All study participants or their guardians provided informed consent for participation in the CIBMTR Research Database and Research Sample Repository Protocols (NCT01166009 and NCT00495300). Non-controlled access data were obtained from public resources – 1000 Genomes Project and GTEx.

### Human samples used for targeted PacBio-seq and TaqMan genotyping of select SNPs

Publicly available DNA samples for HapMap I (CEU panel for CEPH Utah residents with ancestry from Northern and Western Europe, $n = 90$), HapMap III (YRI panel for Yoruba in Ibadan, Nigeria, $n = 90$), select samples from the Human Pangenome Reference Consortium (HPRC, $n = 10$), and the European panel of the Georgia Centenarian Collection ($n = 100$) were purchased from the Coriell Institute for Medical Research. Deidentified tissue samples for bladder tumors and matching adjacent normal samples ($n = 5$ pairs) were purchased from Asterand Bioscience after approval by the NIH Office of Human Subjects Research (#4715) and used for DNA extraction and genotyping. Flow-FISH telomere length samples ($n = 77$) were obtained from donors of hematopoietic cell transplants from the Center for International Blood and Marrow Transplant Research (CIBMTR; https://cibmtr.org) biorepository, comprising 28 females (36.36%) and 49 males (63.64%), ages 21–52 years, mean age 37.68 years. All study participants or their guardians provided informed consent for participation in the CIBMTR Research Database and Research Sample Repository Protocols (NCT01166009 and NCT00495300). The use of the data was approved by the National Marrow Donor Program Institutional Review Board.

Telomere length was measured for total lymphocytes and lymphocyte subsets via the flow-FISH assay described in a previous study[38]. For the current analysis, the samples were selected to represent a wide range of telomeres (4.5–11.2 kb), and telomere length was analyzed in relation to *TERT* genetic variants using linear regression models adjusted for age and sex.

**Cell lines:** The urinary bladder cell lines UMUC3 (CRL-1749), 5637 (HTB-9), HT1376 (CRL-1472), RT4 (HTB-2), T24 (HTB-4), and SCaBER

(HTB-3), as well as the Burkitt lymphoma cell line Raji (CCL-86) and the lung cancer cell line A549 (CCL-185), were purchased from ATCC (Manassas) and maintained in the recommended media supplemented with 10% FBS (unless specified otherwise) and 1% antibiotics. All the cell lines were regularly tested for Mycoplasma contamination using the MycoAlert Mycoplasma Detection Kit (Lonza) and authenticated with the AmpFLSTR Identifiler Plus Kit (Thermo Fisher) if used longer than one year after initial purchase from ATCC. Two versions of EMEM and F-12K complete media for culturing UMUC3 and A549, respectively, were used for xCELLigence, CFSE assay, and apoptosis assay: (1) EMEM or F-12K, both with phenol red, supplemented with 10% FBS and 1% antibiotics (full medium); and (2) phenol red-free EMEM or F-12K supplemented with 10% charcoal-stripped (CS) FBS and 1% antibiotics (CS medium). Cells were moved to CS medium 3–4 days prior to the experiments.

### Analyses of BL tumors

RNA-seq and DNA-WGS data (Illumina) for Burkitt lymphoma (BL) tumors were obtained from the National Cancer Institute (NCI) Cancer Genome Characterization Initiative (CGCI): Burkitt Lymphoma Genome Sequencing Project (BLGSP)[23,58], dbGaP phs000527.v6.p2, including 78 participants (35.90% females and 64.10% males, ages 1–15, mean age 6.95 years). The datasets were accessed through the National Cancer Institute Genomic Data Commons (GDC, https://gdc.cancer.gov/). The RNA-seq BAM files were analyzed using read counts based on the R package Feature-Counts (v2.0.6). Splicing events between *TERT* exons 4 and 5 were annotated based on a custom GTF annotation file to perform read summarization at the feature level, generating a raw count matrix. The total number of reads was determined by counting the reads mapped to the splicing junction between exons 4 and 5 and those that extended into intron 4 by at least 20 bp. Read counts were calculated for the splicing events *INS1* (a 38-bp extension of exon 4 into intron 4), *INS1b* (a 480 bp extension of exon 4 into intron 4), and unspliced intron 4 (total reads between exons 4 and 5 minus reads for *INS1* and *INS1b*) as fractions of the read counts for these events within total read counts. BAM files were also used to estimate the overall expression of *TERT* isoforms −α, −β, and −α−β, which were indexed in a GTF file from ENSEMBL and analyzed using MISO (v0.5.4) with default parameters. Transcripts per million (TPM) values for bulk *TERT* RNA-seq data were downloaded from the GDC data portal. eQTL analyses were performed under additive genetic models using the 'lm' function in R (v4.3.0), with adjustments for sex and age. *TERT* intron retention was analyzed with IRFinder (v2.0.1) with default settings using the GRCh38 reference genome FASTA file and transcriptome GTF file for annotation.

### Analysis of long-read sequences

VNTR6-1 and VNTR6-2 within *TERT* intron 6 were explored based on long-read sequencing data. Phased genome assemblies for 47 individuals (94 chromosomes) were downloaded in FASTA format from the Human Pangenome Reference Consortium (HPRC)[11]. In addition, we used 358 long-read sequencing (R9, Oxford Nanopore) DNA assemblies generated by the Center for Alzheimer's and Related Dementias (CARD) of the National Institute on Aging (available from dbGaP phs001300.v4.p1)[18]. Input DNA was extracted from the brain tissue of 179 neurologically normal individuals of European ancestry and phased assemblies were generated using the Napu pipeline[59].

Genomic sequences in FASTA format were extracted from the assemblies using Cutadapt (v4.0) based on two sets of nested sequences flanking the region of interest, ~9 kb, GRCh38, chr5:1,271,950-1,281,050. The extracted sequences were aligned to the GRCh38 reference genome using minimap2 (v2.26) and combined into one BAM file, with each individual represented by two sequences, one

for each chromosome. In this BAM file, SNPs were scored using SAM-tools with mpileup flag (v1.17), and VNTRs were scored using Straglr (v1.4) with default settings. The pipeline is available at https://github.com/oflorez/HumanGenomeAssemblies and in the repository https://doi.org/10.5281/zenodo.14633198.

## Targeted PacBio-seq

PCR amplicons for targeted PacBio sequencing of VNTR6-1 were generated using the LA Taq Hot-Start DNA Polymerase Kit (Takara) and the M13-tagged primers VNTR6-1-M13F and VNTR6-1-M13R (Supplementary Data 19). In the reference human genome, these primers capture a genomic fragment of 2,241 bp. The optimized 20 μl reactions included 4% DMSO, 0.3 μl of LA Taq DNA Polymerase, 2.5 μl of 10x LA Taq PCR Buffer, 4 μl of 2.5 mM dNTPs, 0.5 μl of each 10 μM primer, and 25 ng of genomic DNA. The PCR conditions included denaturation for 1 min at 94 °C, 36 cycles of denaturation for 10 s at 98 °C and combined annealing/extension for 3.5 min at 68 °C, followed by a final extension for 10 min at 72 °C.

The controls included 1000 G DNA samples purchased from the Coriell Institute for Medical Research and selected to represent various repeat lengths determined based on HPRC assemblies (HG00741, HG01358, HG01891, HG02080, HG02622, HG02717, HG02723, HG03453, HG03492, and NA18906, Supplementary Data 2). For technical validation of the first and second rounds of PCR, all products were quantified with the Quant-iT PicoGreen dsDNA Assay (Invitrogen), and 5% of the products were analyzed with the TapeStation D5000 Kit (Agilent). The second round of PCR was performed with the LA Taq Hot-Start DNA Polymerase Kit and the SMRTbell Barcoded Adapter Complete Prep Kit (PacBio), and the M13 tags incorporated by the first PCR were used to attach unique barcodes to each sample with primers M13F and M13R, where "N" represents the unique barcode (Supplementary Data 19). The 25 μl PCRs included 4% DMSO, 0.4 μl of LA Taq DNA Polymerase, 2.5 μl of 10x LA Taq PCR Buffer II, 4 μl of 2.5 mM dNTPs, 1.0 μl of each 3 μM barcoded M13 primer, and 25 ng of product from the first PCR. The PCR conditions included denaturation for 1 min at 94 °C, 10 cycles of denaturation for 10 s at 98 °C and combined annealing/extension for 3.5 min at 68 °C, followed by a final extension for 10 min at 72 °C. The final amplicons from three 96-well PCR plates (288 samples) were pooled, processed with the Sequel II binding kit 3.1 (PacBio), and sequenced on one SMRT Cell on the Sequel II System (PacBio).

## PacBio amplicon analysis

The high-fidelity (HiFi) reads were assembled by circular consensus sequencing (CCS) within SMRT Link (PacBio), demultiplexed with Lima, and aligned to the reference genome GRCh38 with minimap2. The VNTR6-1 amplicons had an average read coverage of ~10,000 reads per sample. The resulting BAM files were scored for rs56345976 and rs33961405 using SAMtools with mpileup flag (v1.17) and for VNTR6-1 using Straglr (v1.4). The analysis was restricted by reads fully covering the amplicon (GRCh38, chr5:1275500-1277500), excluding outputs from partial reads using SAMtools with the ampliconclip flag (v1.17). Phased haplotypes of rs56345976 and rs33961405 were constructed based on PacBio reads.

## DNA genotyping

TaqMan genotyping assays for the *TERT* SNPs rs56345976 (C_88 595060_10), rs33961405 (C_34209972_10), rs10069690 (C_3032 2061_10), rs2242652 (C_16174622_20), rs7705526 (C_189441058_10), rs2736100 (C_1844009_10) and rs2853677 (C_1844008_10) were purchased from Thermo Fisher. The samples were genotyped in 384-well plates on a QuantStudio 7 Flex Real-Time PCR System (Applied Biosystems) using 2x TaqMan Genotyping Master Mix (Thermo Fisher) in 5-μL reactions with 4 ng of genomic DNA per reaction.

## Analyses in the 1000 Genomes (1000 G) Project

High-coverage (30x) short-read WGS data in CRAM format and phased genetic variants for 3201 individuals from the 1000 G populations[60] were downloaded from https://www.internationalgenome.org/data-portal/data-collection/30x-grch38 for the 400 kb genomic region (GRCh38 chr5:1,100,000-1,500,000). The depth of coverage of the aligned short-sequencing reads within the 2290 bp genomic region corresponding to VNTR6-1 (GRCh38 chr5:1,275,210-1,277,500) was analyzed by calculating the median coverage within consecutive 50-base windows using Mosdepth (v0.2.5). All the samples were classified into VNTR6-1-Short/Short genotypes (24–27 copies) and Long/any genotypes (with one or two Long alleles of 40.5 or 66.5 copies) by applying a machine learning approach based on regularized multi-modal logistic regression, which was developed with the tidymodels framework and the R package 'glmnet' (v4.1-7). First, a total of 605 samples (18.89%) were randomly selected from the set, representing all 1000 G super-populations, and visually examined and assigned to the Short or Long groups based on the coverage profiles in IGV. The dataset was then split into training (60%) and testing (40%) sets. Fivefold cross-validation was used during the training process to develop and evaluate the prediction model. The model demonstrated stable performance in accurately classifying VNTR6-1 into the Short and Long categories, with 96.8% specificity, 92.8% sensitivity, an F score of 0.95, and an area under the ROC curve (AUC) of 0.98 (Supplementary Fig. 2).

To identify variants predictive of VNTR6-1-Short/Long status, all 12,338 biallelic SNPs from the 1000 G phased genetic variant data across the 400 kb genomic region (GRCh38 chr5:1,100,000-1,500,000) were extracted and filtered for an MAF > 5%, resulting in 1473 SNPs for analysis. Based on Chi-square tests, 594 of these SNPs were significantly associated with the VNTR6-1 Short and Long categories ($p < 0.05$). A random forest model was then applied using the R package 'randomForest' (v4.7-1.1) to identify the predictive value of the significant SNPs for VNTR6-1 categories, selecting the top 10% based on mean decrease in Gini scores. A total of 60 SNPs were identified as highly informative, with rs56345976 and rs33961405 showing the highest combined predictive probabilities for VNTR6-1 classification.

To map the haplotypes of rs56345976 and rs33961405 with the profile of coverage distribution across the genomic region GRCh38 chr5:1,275,210-1,277,500, we applied unsupervised hierarchical clustering using the core 'hclust' function in R (v4.3.0) with the Euclidean distance metric and complete linkage method. The rs56345976-A/rs33961405-G haplotypes captured the VNTR6-1 Long allele (Cohen's Kappa coefficient of 0.78 and agreement of 0.90), whereas all the remaining haplotypes captured the VNTR6-1 Short allele (Supplementary Fig. 4). Phased data from our long-read sequencing, including assemblies and targeted sequencing, were used to confirm the segregation of rs56345976 and rs33961405 with VNTR6-1 (Supplementary Data 2).

We created a custom 1000 G reference panel that included all the markers within the 400 kb genomic region (GRCh38 chr5:1,100,000-1,500,000). In this region, VNTR6-1 was used as a biallelic marker, with Short and Long alleles determined by the rs56345976/rs33961405 haplotypes at position chr5:1,275,400 (Supplementary Data 3). To evaluate the scoring performance, the 1000 G dataset ($n = 3201$) was randomly partitioned into two groups, which served as a reference panel ($n = 1601$) and a test panel ($n = 1600$), to perform phasing with SHAPEIT4 (v4.2.0) and imputation with IMPUTE2 (v2.3.2) with default settings. VNTR6-1 was confidently scored in all test panel samples (imputation quality score[61], IQS = 0.98), with an overall concordance of 99.3% compared with the predetermined genotypes across the entire dataset. The population-specific concordance rates for VNTR6-1 imputation were as follows: EUR 99.7% ($n = 321$), AMR 99.6% ($n = 243$), AFR 99.1% ($n = 456$), SAS 98.99% ($n = 299$), and EAS 98.32% ($n = 281$).

## Analyses in the Prostate, Lung, Colorectal and Ovarian (PLCO) Cancer Screening Trial

PLCO[62] is a large population-based cohort that includes 155,000 participants enrolled between November 1993 and July 2001. The individual-level data, including genotyped variants from Illumina arrays, imputed variants using the TopMed reference panel, and phenotype data, were provided by PLCO upon approved application (project #PLCO-957). The European ancestry dataset included 99,167 individuals (51.66% females and 48.34% males, ages 42–74, mean age 62.26 years), comprising 73,085 cancer-free controls (55.28% females and 44.72% males, mean age 62.02 years) and 26,082 patients with 16 cancer types (41.52% females and 58.48% males, mean age 62.84 years), 3239 (12.42%) of whom had multiple cancer types. All the variants within the 400 kb region (GRCh38 chr5:1,100,000-1,500,000) were phased using SHAPEIT4 (v4.2.0) and then VNTR6-1 genotypes (Short or Long) were assigned based on phased rs56345976/rs33961405 haplotypes. Logistic regression analyses were conducted with the logit link function for binary outcomes using the 'glm' function in R (v4.3.0), adjusting for sex and age.

## Analyses in the UK BioBank

Associations between genetic markers and relative leukocyte telomere length (rLTL) in peripheral blood were assessed in the UK Biobank (UKB) (https://www.ukbiobank.ac.uk/), a population-based prospective study in the United Kingdom[63], based on an approved application (#92005). The analysis included 339,103 cancer-free participants of European ancestry (54.64% females and 45.36% males, ages 38–73, mean age 55.82 years) with SNP data genotyped using the UK Biobank Axiom array and imputed using the Haplotype Reference Consortium and UK10K reference panels, along with rLTL measurements. VNTR6-1 was scored as described above for PLCO. We used linear regression models to assess the associations between the technically adjusted rLTLs ($\log_e$ and Z-transformed)[64] and the genetic markers. This analysis was performed using the 'lm' function in R (v4.3.0) and adjusting for sex, age, and smoking status. A conditional linear model was tested by independently adding SNPs (rs2736100, rs2853677, and rs7705526) that are strongly associated with telomere length in multiple populations. To account for trend differences in rLTLs across all ages, the conditional linear model included an interaction term between the genetic markers and 5-year age groups that was used to avoid age-heaping bias while maintaining a sufficient sample size for each age class.

## Analyses in The Cancer Genome Atlas (TCGA)

Blood-derived germline data for 9,610 TCGA participants across 33 cancer types were accessed through the National Cancer Institute Genomic Data Commons (GDC, https://gdc.cancer.gov/). Controlled access genotype calls generated from Affymetrix SNP6.0 array intensities using BIRDSUITE[65] were retrieved from the genomic region GRCh37, chr5:335,889-2,321,650. In this region, in addition to the 5453 initially genotyped variants, we imputed approximately 57,000 variants with imputation quality scores exceeding 0.8 using the TOPMed Imputation Server, which includes data from more than 97,000 participants[66]. The imputation quality scores across cancer types were as follows: mean (min–max) $r^2 = 0.83$ (0.78-0.89) for rs56345976, $r^2 = 0.85$ (0.75-0.92) for rs33961405, $r^2 = 0.85$ (0.76-0.94) for rs10069690, and $r^2 = 0.84$ (0.74-0.92) for rs2242652. Direct genotyping from germline WGS files for 387 BLCA downloaded from the GDC revealed high concordance rates between imputed and WGS-genotyped markers: 89.90% for rs56345976, 86.79% for rs33961405, 91.17% for rs10069690 and 92.75% for rs2242652.

Transcripts per million (TPM) for bulk *TERT* RNA-seq data were downloaded from the GDC within the Pan-Cancer Atlas publications[67]. The TPMs for the *TERT*-β and *TERT*-FL transcripts were downloaded from the UCSC Xena platform (https://xenabrowser.net/datapages/)

within the UCSC toil RNA-seq Recompute Compendium, cohort TCGA Pan-Cancer (PANCAN). We used pre-computed telomerase-related metrics, including expression-based telomerase enzymatic activity detection (EXTEND) scores based on a 13-gene signature[33], stemness indices calculated via a predictive model using one-class logistic regression on mRNA expression[34], a telomerase signature score estimated from a 43-gene panel, and telomere length scores calculated using TelSeq based on WGS[35].

eQTL analysis was conducted using TPMs for bulk RNA-seq *TERT* expression data and genetic markers (additive genetic model) using the 'lm' function in R (v4.3.0), with adjustments for sex and age. Spearman rank correlations between *TERT* expression (*TERT*-β and *TERT*-FL) and telomerase-associated metrics for each cancer type were determined using the 'rcorr' function of the Hmisc package in R (v4.3.0).

## Analyses in the Genotype-Tissue Expression (GTEx) project

TPMs for the *TERT*-β and *TERT*-FL transcripts were downloaded from the GTEx Portal (https://gtexportal.org/home/downloads/) within the bulk tissue expression database, GTEx Analysis V8 RNA-seq. Pre-computed EXTEND scores based on a 13-gene signature were obtained from the Supplementary Information of the corresponding publication[33]. Spearman rank correlations between *TERT* expression (*TERT-β* and *TERT*-FL) and EXTEND scores for each tissue type were determined using the 'rcorr' function of the Hmisc package in R (v4.3.0). The eQTLs for rs10069690, rs2242652, and *TERT* expression were assessed through the GTEx portal.

## CFSE proliferation assay

For each condition, cells (9.6E5) were stained with a 5 μM solution of carboxyfluorescein succinimidyl ester (CFSE) dye (CellTrace CFSE Cell Proliferation Kit, Thermo Fisher) for 15 min at 37 °C. Culture media containing 10% CS FBS was added to an equal volume of staining solution to quench excess dye. CFSE-stained cells were seeded into 6-well plates (Corning) at 1.2E5 cells/well in CS medium and incubated at 37 °C and 5% $CO_2$. The remaining CFSE-stained cells were analyzed on an AttuneNxT (Thermo Fisher) flow cytometer to determine the day 0 (maximal) CFSE intensity ($CFSE_{start}$). Seeded cells were grown for 48 h in CS medium to allow all cell lines to reach a sufficient level of attachment for a medium change and then switched to either full medium or CS medium. The cells were harvested with 0.05% trypsin-EDTA 48 h after the media were changed and analyzed by flow cytometry to determine the final CFSE intensities ($CFSE_{final}$). The data were re-analyzed using FlowJo v10. The CFSE mean fluorescence intensity (MFI) was determined by taking the geometric mean of fluorescence (collected on the BL1 channel, 530/30 nm) after gating live single cells. Cell doublings were calculated using the equation: Cell Doublings = − (ln ($CFSE_{final}$/$CFSE_{start}$)/ln 2).

## CRISPR/Cas9 genome editing

CRISPR/Cas9 guide RNAs flanking the VNTR6-1 region (2241 bp in the reference genome) were designed using sgRNA Scorer 2.0[68]. Annealed oligonucleotides corresponding to two guide RNAs (Supplementary Data 19) were cloned using Golden Gate Assembly cloning into PDG458 (ref. 69, Addgene plasmid #100900; http://n2t.net/addgene:100900; RRID:Addgene 100900, a gift from Paul Thomas). The cells (1.0E6/transfection) were transiently transfected with CRISPR/Cas9-expressing plasmids using the Amaxa 4D nucleofection system (Lonza), a 100 μl SF cell line kit, and the CM-130 program (A549 profile settings were used for all the cell lines). GFP-positive cells were enriched by FACS 48 h post-transfection using an SH800 sorter (Sony). The enriched population was further single-cell sorted in 96-well plates to isolate pure knockout populations. Genomic DNA from the expanded clones was screened by PCR with the primers VNTR6-1F and VNTR6-1R (Supplementary Data 19). These primers generate a 2241 bp PCR

product (based on the reference genome sequence) and a 974 bp PCR product after knockout. Three independent knockout clones (V6.1-KOs) were selected for functional analyses. Clones that were exposed to CRISPR reagents but did not result in knockout were compared with parental controls (WT, no CRISPR treatment) by RNA-seq analysis. CRISPR treatment had negligible effects on gene expression, and statistical analysis of the RNA-seq data was performed comparing V6.1-KOs with the WT.

## Cloning

The pCMV-TERT-FL-HA expression construct was generated with high-fidelity Q5 polymerase (NEB) and amplified from a *TERT-FL* plasmid (GenScript OHu25394), using a forward primer with an AgeI recognition site and a reverse primer with an HA-tag and BsrGI recognition sites (Supplementary Data 19). PCR fragments were isolated by electrophoresis and a gel extraction kit (Qiagen) and cloned into an mEGFP-N1 expression vector (Addgene #54767) using AgeI and BsrGI restriction enzymes (NEB), replacing mEGFP. The pCMV-TERT-β-3xFLAG expression construct was generated using two separate Q5 PCRs from the same *TERT*-FL plasmid. The first PCR utilized the same AgeI forward primer and a reverse primer with a native BamHI recognition site within *TERT* exon 9. The second PCR utilized the BamHI site in its forward primer and a reverse primer with a 3xFLAG tag and a BsrGI recognition site (Supplementary Data 19). These two PCR fragments were isolated by electrophoresis and a gel extraction kit (Qiagen), cloned into pCR4 Blunt-TOPO (Invitrogen), and subcloned into the mEGFP-N1 expression vector using AgeI + BamHI and BamHI + BsrGI, replacing mEGFP.

## RNA extraction

Cell lysates were harvested from culture plates using 350 µl of RLT lysis buffer/well and stored at −80 °C before extraction. RNA was extracted with the Qiagen RNeasy Mini RNA kit using QIAcube with standard on-column DNAse treatment (Qiagen). The RNA concentrations were quantified with a Qubit RNA High Sensitivity Kit (Invitrogen).

## cDNA synthesis

7.5 µg of RNA from each sample was used in 20 µl reactions with the iScript Advanced cDNA Synthesis Kit (Bio-Rad). The cDNA was concentrated overnight by ethanol precipitation and resuspended in 37.5 µl of water, resulting in an RNA input concentration of 200 ng/µl.

## Expression assays

Expression of the *TERT-β* and *TERT-FL* transcripts was quantified with two custom TaqMan gene expression assays (Thermo Fisher, Supplementary Data 19) designed to target specific exons and splice junctions. Reactions were multiplexed to include both targets and a custom human *HPRT1* endogenous control (NED/MGB probe, primer limited, Assay ID: Hs99999909_m1, Thermo Fisher). TaqMan reactions were run in technical quadruplicate in 384-well plates on a Quant-Studio 7 Flex Real-Time PCR System (Applied Biosystems). Each 6 µl reaction included 2 µl of cDNA diluted to 100 ng/µl from a 200 ng/µl RNA input. All assays (individually and in multiplexed reactions) were validated using the *TERT*-FL-HA and *TERT*-β-3xFLAG plasmids in a 5 × 10-fold dilution series (from 100 pM to 10 fM). All the assays had experimentally determined PCR efficiencies ranging from 72–100%. The identities of the PCR products were confirmed by cloning into the TOPO-pCR4 vector (Invitrogen) and Sanger sequencing with the M13_TOPO primers (Supplementary Data 19).

SYBR Green RT–qPCR assays were performed with iTaq Universal SYBR Green Supermix (Bio-Rad). The samples were run in 5 µl reactions with 2 µl of cDNA diluted to 50 ng/µl from the RNA input in 12 technical replicates on a QuantStudio 7 Flex Real-Time PCR System. The primers (10 mM, Thermo Fisher) used were identical to those used in the

TaqMan assays. *HPRT1* controls (Supplementary Data 19) were run in parallel reactions. For visualization, technical replicates of selected RT–qPCR products were pooled and resolved on 2% agarose gels, along with a low-molecular-weight DNA ladder (NEB). The gel images were captured on a Bio-Rad ChemiDoc Imaging System and analyzed using Image Lab Software v6.1.0 (Bio-Rad). The ratio of *TERT* isoforms was calculated based on the gel densitometry of the PCR products (120 bp and 302 bp).

Total *TERT* expression was measured in 5 µL reactions using TaqMan assays (FAM, exons 3-4) with *TERT*-Hs00972650_m1 multiplexed with the endogenous control *HPRT1* (VIC, primer-limited, Assay ID: Hs99999909_m1) and TaqMan Gene Expression Buffer (all from Thermo Fisher).

## RNA-seq

RNA quality (all RINs>9.0) was verified using the Bioanalyzer (Agilent) and an RNA 6000 Nano Kit (Agilent). For each sample, 200 ng of total RNA was used to prepare an adapter-ligated library with the KAPA RNA HyperPrep kit with RiboErase (HMR) (KAPA Biosystems) using the xGen Dual Index UMI Adapters (IDT). The multiplexed libraries with 250–350 bp inserts were sequenced on a NovaSeq 6000 (Illumina) to generate 279–418 million paired-end 150 bp reads per sample. Quality assessment of the RNA-seq data was conducted using MultiQC (v1.16)[70]. The quantification of transcript abundance was performed using Salmon (v0.14.1) in count mode with−validateMappings flag and expressed as transcripts per million (TPM). The raw RNA-sequencing reads were aligned with STAR[71] based on the reference genome GRCh38 and GENCODE annotation (v36). Differential expression analysis was conducted with DESeq2 (v1.40.2) based on the estimated counts obtained from Salmon quantification, controlling for the false discovery rate (FDR). Gene-level transcript abundances were estimated with 'lengthScaledTPM' in the R package 'tximport' (v1.28.0). Gene Ontology (GO) analysis and gene set enrichment analysis (GSEA) on differentially expressed genes was conducted with clusterPro-filer (v4.8.3).

## G4 Hunter prediction analysis

Analysis was performed with G4Hunter (https://bioinformatics.ibp.cz)[72]. PacBio-generated DNA sequences for UMUC3 (24 repeat copies per allele) and HG03516 (27 and 66.5 repeat copies per allele) were used as inputs flanked by 120 bp on each side of the VNTR6-1 region.

## G4-seq analysis

For the lymphoblastoid cell line NA18057 (VNTR6-1-Short/Short genotype, 24 and 27 repeat copies), ChIP-seq data for G-quadruplexes (G4) detected in forward and reverse orientations were downloaded from BED files from the GEO dataset GSE63874 (ref. 73, files GSE63874_Na_K_minus_hits_intersect.bed.gz and GSE63874_Na_K_plus_hits_intersect.bed.gz). These files were merged into a single BED file and converted to the UCSC BED format. The G4 mismatch quantification bedGraph files GSE63874_Na_K_12_minus.bedGraph.gz and GSE63874_Na_K_12_plus.bedGraph.gz were downloaded and converted into bigwig format using the bedGraphToBigWig tool.

Similarly, for the 293 T normal embryonic kidney cell line (VNTR6-1-Long/Long genotype), the G4-seq data were downloaded from GSE110582 (ref. 74, files GSM3003539_Homo_all_w15_th-1_minus.hits.max.K.w50.25.bed.gz and GSM3003539_Homo_all_w15_th-1_plus.hits.max.K.w50.25.bed.gz) and processed as above. The G4 mismatch quantification values were downloaded from GSM3003539_Homo_all_w15_th-1_minus.K.bedGraph.gz and GSM3003539_Homo_all_w15_th-1_plus.K.bedGraph.gz. The G4-seq tracks for NA18057 and 293 T cells were visualized through the UCSC Genome Browser (GRCh37).

## Evaluation of G4 ligands

Five G4 stabilizing ligands were tested for their ability to stabilize TERT G4. Ligands: PhenDC3, TMPyP4, BRACO-19, and Pyridostatin were provided by Dr. John Schneekloth. Pidnarulex (CX-5461) was selected from the literature[21] and obtained from Selleck Chem. For optimization, cells were seeded into 6-well plates at 4.0E5 cells/well. After adhering for 24 h, the cells were treated for 24, 48, or 72 h with 0.1 μM, 0.3 μM, 1 μM, 3 μM, 10 μM, or 30 μM ligands dissolved in DMSO, with the DMSO vehicle alone and untreated control samples included in each plate. In the 72 h group, the media was replaced at 48 h, and the cells were harvested at 72 h. The viability of the treated cells was evaluated by counting them with the BioTek Lionheart FX automated microscope (Agilent) every 24 hours. Pidnarulex (CX-5461) and PhenDC3 at 3 μM for 72 h were identified as the most effective treatments for modulating *TERT* exon 7-8 skipping and were used in subsequent experiments. WT and V6.1-KO cells were treated with technical replicates in three independent experiments.

## Western blot

BCA-normalized protein samples and 10 μL of SeeBlue Plus2 ladder were loaded and run on gels using 1X Bolt running buffer at 165 V for 1 h and transferred to nitrocellulose membranes using an iBlot2 dry transfer instrument (Invitrogen). The membranes were blocked with 5% milk in 1X TBST for 1 h at room temperature. The membranes were incubated overnight at 4 °C with primary antibodies in 2.5% milk in 1× TBST (anti-GFP: Invitrogen A-11122; anti-HA: Novus NB600-362; anti-FLAG: Sigma–Aldrich M2; anti-GAPDH: Abcam ab9485). After three 5 min washes with 1X TBST, the membranes were incubated at room temperature for 1 h with secondary antibodies (anti-rabbit: Cell Signaling 7074; anti-mouse: Cell Signaling 7076; anti-goat: Santa Cruz sc-2304) and imaged using Pico and Femto ECL reagents (Thermo).

## Structured illumination microscopy fluorescence imaging

A549 cells were chosen for imaging of mitochondria because this highly transfectable cell line has a larger cytoplasmic area than UMUC3, allowing better visualization. The cells were seeded in a 12-well plate at 1.25E5 cells/well and cotransfected with pCMV-TERT-FL-HA or pCMV-TERT-β-3xFLAG expression constructs at a 50:50% isoform ratio. Transfections were performed using Lipofectamine 3000 for 4 h. The transfected cells were washed with DPBS, dissociated using Accutase (StemPro), and counted. The cells were then diluted and seeded onto CultureWell Chambered Coverglass (Invitrogen). After 48 h, the coverslips were fixed with 4% formaldehyde in PBS for 10 min, permeabilized with 0.03% Triton-X 100 for 10 min, and blocked with blocking buffer (5% BSA + 0.01% Triton-X 100 in PBS) for 30 min. The coverslips were incubated at 4 °C overnight with the following primary antibodies: anti-FLAG (Sigma M2, mouse, 1:400 dilution), anti-HA (Novus NB600-362, goat, 1:400 dilution), and anti-TOM20 (Proteintech 11802-1-AP, rabbit, 1:1000 dilution) diluted in blocking buffer, followed by incubation at room temperature for 30 min with the following secondary antibodies: anti-mouse-AlexaFluor488 (Thermo Fisher A21202, 1:500 dilution), anti-goat-AlexaFluor647 (Thermo Fisher A32849, 1:500 dilution), and anti-rabbit-AlexaFluor555 (Thermo Fisher A31572, 1:500 dilution) diluted in blocking buffer. Three washes were performed with PBS between all the staining steps; after the final wash, the cells were counterstained with 3 μg/ml DAPI. The coverslips were then mounted onto glass slides with ProLong Gold Antifade Mountant (Invitrogen) and sealed with clear nail polish. Superresolution structured illumination microscopy fluorescence images were obtained using ZEN Black software on an ELYRA PS.1 A superresolution (SR) microscope (Carl Zeiss, Inc.) with a Plan-Achromat 63X/1.4 NA oil objective and a Pco.edge sCMOS camera, 405 nm/488 nm/561 nm/633 nm laser illumination and standard excitation and emission filter sets were used. Raw data were acquired by projecting grids onto the sample

generated from the interference from a phase grating with 23 μm, 28 μm, and 34 μm spacings for 405, 488, and 561 nm excitation, respectively (3 grid rotations and 5 grid shifts for a total of 15 images per super-resolved z-plane per color). The raw images were processed with ZEN black software. For publication, images were scaled to 8-bit RGB identically with a linear LUT and exported in TIFF format using ImageJ. Figures were made from the TIFF images in Adobe Illustrator without any change in resolution, except for the inset zoomed images.

## Apoptosis assay

Cells were seeded in 6-well plates at 1.2E5 cells/well, and the media was changed 48 h later to full medium, CS medium alone, or CS medium containing 10 μM cisplatin. The cells were harvested with 0.05% trypsin-EDTA 48 h after the media was changed, pelleted at 500 × g for 5 min, and washed with 1 mL of PBS. The cells were stained with an Annexin V-FITC conjugate (Thermo Fisher) and propidium iodide (Thermo Fisher) in Annexin V staining buffer (Thermo Fisher) according to Rieger et al.[75]. FITC (ex.488 nm/em.517 nm) and PI (ex.488 nm/em. 617 nm) fluorescence were analyzed by flow cytometry on an Attune NxT with a CytKick Autosampler (Thermo Fisher). Unstained cells, Annexin V-FITC-stained cells, and PI-stained cells were used as compensation controls. Apoptosis was determined by the percentage of FITC-positive cells.

## Lionheart cell proliferation analysis

WT and V6.1-KO UMUC3 cells were seeded in 6-well plates (Falcon) at 1.0E4 cells/well in EMEM. After adherence for 24 hours (day 0), a label-free cell counting protocol, with focus and cell size calibrated to adhered WT UMUC3 cells, was created on the BioTek Lionheart FX automated microscope (Agilent), and cell counts were recorded every 24 h for 10 days. The fold change in the number of cells was calculated by dividing the recorded counts by the initial cell counts on day 0.

Linear mixed models were applied to the data obtained from the Lionheart FX, normalized to day 0, where the treatment type was considered a fixed effect term and the technical replicate was considered a random effect term. Maximum likelihood estimation procedures were employed to conduct joint effects likelihood-ratio tests, whereas restricted maximum likelihood estimation was utilized for more precise estimation of effect sizes as beta coefficients using the linear mixed-effects function in the R package 'nlme' (v3.1–162).

## xCELLigence Real-Time Cell Analysis (RTCA)

In Supplementary Fig. 19c and d, cells were seeded in a 12-well plate at 1.25E5 cells/well and transfected with either GFP, pCMV-TERT-FL-HA, or pCMV-TERT-β-3xFLAG expression constructs either as single transfection or co-transfection at different ratios of isoforms (80:20% and 20:80%). Transfections were performed using Lipofectamine 3000 for 4 h. The transfected cells were washed with DPBS, dissociated using Accutase (StemPro), and counted. The cells were then diluted and seeded into an xCELLigence E-Plate 16 microplate (Agilent) at 1.0E3 cells/well and placed on an xCELLigence RTCA DP system (Agilent). The data were collected every 15 minutes in RTCA software for 288 h and then exported for analysis.

In Fig. 4a and Supplementary Figure 18, 1.0E3 WT or V6.1-KO cells grown in CS medium were seeded into E-Plate 16 microplates at 1.0E3 cells/well (Agilent). Cell label-free impedance in the E-Plate (correlated with cell proliferation) was measured every 15 min for 283 h using the xCELLigence RTCA DP system. Two days after seeding, the medium was changed to either full medium or CS medium (control).

Linear mixed models were applied to the impedance data obtained from the xCELLigence system, where the treatment type was considered a fixed effect term and the technical replicate was considered a random effect term. Maximum likelihood estimation procedures were employed to conduct joint effects likelihood-ratio tests,

whereas restricted maximum likelihood estimation was utilized for more precise estimation of effect sizes as beta coefficients using the linear mixed-effects function in the R package 'nlme' (v3.1–162).

## HiChIP analysis

The H3K27Ac HiChIP libraries for the bladder cancer cell lines T24 and RT4 were generated using the Arima-HiChIP protocol (Arima Genomics, A101020). Briefly, 1E6 cells/replicate were collected for chromatin cross-linking followed by digestion with a restriction enzyme cocktail, biotin labeling, and ligation. The samples were then purified, fragmented, and enriched. Pulldown was performed using an antibody against H3K27ac (Cell Signaling Technology, #8173). The Arima-HiChIP libraries that passed the QC were sequenced using an Illumina NovaSeq 6000 to generate raw FASTQ files for each sample. The paired-end reads were aligned to the GRCh37 genome using the HiC-Pro pipeline (v3.1.0, https://github.com/nservant/HiC-Pro). The confirmed interaction reads were used as inputs for significant loop calling via the FitHiChIP tool (v.11.0, https://github.com/ay-lab/FitHiChIP) with default settings. The HiChIP loop and ATAC peak calling files for the GM12878 and normal bladder samples were downloaded from the Gene Expression Omnibus (GSE188401). The interactions were visualized through the UCSC genome browser.

## PacBio DNA methylation analysis

Freshly collected genomic DNA (5 µg) from the HT1376, RT4, T24, SCaBER, UMUC3, and Raji cell lines was sheared using Covaris g-tubes at 4800 rpm, followed by size selection using PippinHT. Three SMRT flow cells were run for each sample library on the PacBio Sequel II platform. The sequence reads were transformed into FASTQ files and aligned to the GRCh38 reference genome using the default settings of the SMRT-Link workflow. 5mC DNA methylation analysis was a part of the SMRT-Link pipeline, and the corresponding information specifying the positions and probabilities of 5mC methylation at CpG sites was integrated into the output file. The methylation plots were generated in IGV by coloring alignments in PacBio WGS bam files based on base modification (5mC).

## Oxford Nanopore cDNA-seq

cDNA libraries were generated using the PCR cDNA Sequencing Kit SQK-DCS109 (Oxford Nanopore Technologies), starting with 100 ng of poly-A RNA. Libraries were loaded onto R9.4.1 PromethION flow cells mounted on a P2 Solo and run for 96 h. Basecalling was performed using MinKNOW software with the high-accuracy model on a GridION sequencer (Oxford Nanopore Technologies). The reads were aligned to GRCh38 via Minimap2 (v2.26) and SAMtools (v1.5). UMUC3 yielded 25,827,200 reads, with 46 reads aligning to *TERT*, whereas UMUC3 V6.1-KO yielded 18,709,848 reads, with 62 reads aligning to *TERT*.

## Analysis of sequence conservation in non-human species

Haplotype-resolved Telomere-to-Telomere (T2T) assemblies of primates were downloaded from GenomeArk (https://www.genomeark. org/). The FASTA sequences were aligned to the human GRCh38 reference genome using Minimap2 (v2.26) with the '-ax asm10' flag and converted to a BAM file using SAMtools (v1.5). The *TERT* VNTR6-1 repeat units were analyzed with Tandem Repeat Finder (https:// tandem.bu.edu/trf/trf.html). The BAM files of Neandertal (*n* = 3) and Denisova (*n* = 1) individuals were downloaded from the Max Planck Institute for Evolutionary Anthropology resource (http://cdna.eva. mpg.de/neandertal/Vindija/bam/Pruefer_etal_2017/ and http://cdna. eva.mpg.de/neandertal/Chagyrskaya/) and visualized using IGV.

## Statistical analysis

Analyses were performed with R Studio (v4.3.0), GraphPad Prism (v10), and FlowJo (v9). P values are for unpaired two-sided tests: Student's *T* test, linear mixed models, and linear or logistic regression, with adjustments for relevant covariates as indicated. *P*-values are reported without correction for multiple comparisons, or based on FDR-adjustment or permutation as indicated. Error bars correspond to standard deviation (SD), standard error of the mean (SEM), or 95% confidence intervals (CI), as indicated.

This work utilized the computational resources of the NIH HPC Biowulf cluster (http://hpc.nih.gov). The figures were assembled using Adobe Illustrator.

## Reporting summary

Further information on research design is available in the Nature Portfolio Reporting Summary linked to this article.

## Data availability

Data generated in this study have been deposited in the NCBI Sequence Read Archive (SRA). PacBio-targeted sequencing data are included in the BioProject PRJNA1134698. The data for PacBio-WGS, HiChIP, short-read RNA-seq by Illumina, and long-read RNA-seq by Oxford Nanopore Technology are included in the BioProject PRJNA1134701. The publicly available datasets used in the study include RNA-seq expression data from TCGA (UCSC Xena platform, https://toil.xenahubs.net, https://toil-xena-hub.s3.us-east-1.amazonaws.com/download/tcga_rsem_isoform_tpm.gz)[76]; RNA-seq expression data in normal tissues (GTEx portal, https://www.gtexportal.org/, https://storage.googleapis.com/adult-gtex/bulk-gex/v8/rna-seq/GTEx_Analysis_2017-06-05_v8_RSEMv1.3.0_transcript_tpm.gct.gz)[77]; haplotype-resolved Telomere-to-Telomere (T2T) assemblies of primates (Genome Ark database, https://www.genomeark.org/, IDs: mGorGor1, mPanPan1, mPanTro3, and mPonAbe1, accessed on October 3, 2023 https://registry.opendata.aws/genomeark); WGS for Neandertal and Denisova individuals (Max Planck Institute for Evolutionary Anthropology, https://www.eva.mpg.de/index/, IDs: Altai, Denisova, Vindija, and Chagyrskaya)[78,79]; FASTA files for long-read WGS (Human Pangenome Reference Consortium, https://humanpangenome. org/, https://github.com/human-pangenomics/HPP_Year1_Data_Freeze_v1.0)[11]; HiChIP data (NCBI Gene Expression Omnibus, accession code GSE188401)[80]; the 1000 Genomes 30x on GRCh38 data (The International Genome Sample Resource, https://www.internationalgenome. org/, https://www.internationalgenome.org/data-portal/data-collection/30x-grch38)[60]; and ChIP-seq data for G-quadruplexes (NCBI Gene Expression Omnibus, accession codes GSE63874[73] and GSE110582)[74]. The controlled access data were obtained from PLCO (#PLCO-957) and UKB (#92005) based on approved applications. The controlled access long-read sequencing data from the Center for Alzheimer's and Related Dementias (CARD) of the National Institute on Aging is available from dbGaP phs001300.v4.p1, and data for Burkitt Lymphoma Genome Sequencing Project (BLGSP) is available from dbGAP phs000527.v6.p2. The remaining data used in this article are available within the Article, Supplementary Information, or Source data provided with this paper. Source data are provided in this paper.

## Code availability

The pipeline and script used for the analysis of the genome assemblies are available at GitHub (https://github.com/oflorez/HumanGenomeAssemblies) and Zenodo (https://doi.org/10.5281/zenodo.14633198)[81].

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

## Acknowledgements

This work was supported by the Intramural Research Programs of the Division of Cancer Epidemiology and Genetics (DCEG) and the Center for Cancer Research (CCR), the National Cancer Institute, and the Center for Alzheimer's and Related Dementias (CARD) within the Intramural Research Program of the National Institute on Aging and the National Institute of Neurological Disorders and Stroke (1ZIAAG000538). BLGSP was funded in part by the Foundation for Burkitt Lymphoma Research (http://www.foundationforburkittlymphoma.org) and with U.S. Federal funds from the National Cancer Institute, National Institutes of Health, under Contract No. HHSN261200800001E and Contracts No. HHSN261201100063C and No. HHSN261201100007I (DCEG). The presented results are, in part based upon data generated by the TCGA Research Network. The work was conducted using the UK Biobank resource (application 92005). The UK Biobank was established by the Wellcome Trust, the Medical Research Council, the United Kingdom Department of Health, and the Scottish Government. The UK Biobank has also received funding from the Welsh Assembly Government, the British Heart Foundation, and Diabetes UK. The CIBMTR is supported primarily by the Public Health Service U24CA076518 from the NCI, the National Heart, Lung and Blood Institute (NHLBI), and the National Institute of Allergy and Infectious Diseases (NIAID); 75R60222C00011 from the Health Resources and Services Administration (HRSA); and N00014-23-1-2057 and N00014-24-1-2057 from the Office of Naval Research. The Cancer Genomics Research (CGR) Laboratory and Genome Modification Core are funded with Federal funds from the National Cancer Institute under Contract No. 75N910D00024. B.P. and M.M. acknowledge the support of the Chan Zuckerberg Initiative and the National Institutes of Health grants U24HG011853 and OT2OD033761 to B.P. M.H.H. was supported by the NCI Intramural Continuing Umbrella for Research Experiences (iCURE) program. We thank Drs. Helen Piontkivska, and the members of the Laboratory of Translational Genomics for comments and discussions. We thank Dr. Tatiana Karpova, Optical Microscopy Core (NCI/CCR/LRBGE), for helping with super-resolution imaging. The opinions expressed by the authors are their own and should not be interpreted as representing the official viewpoint of the

U.S. Department of Health and Human Services, the National Institutes of Health, or the National Cancer Institute. Open Access funding was provided by the National Institutes of Health (NIH).

## Author contributions

O.F.-V. and L.P.-O. conceived the study; O.F.-V., C.-H. L., and C.Z. performed the data analysis; M.H., M.H.H., B.W.P., and K.F. performed the experiments; C.B., K.J.B., M.K., M.M., and B.P. generated the long-read genome assemblies; K.F., M.H.H., K.J., W.L., and K.T. performed the long-read targeted sequencing; R.C., J.S., M.J.M., S.J.C., S.M.G., S.A.S., and S.M.M. provided reagents, data, samples and interpretations of the results; O.F.-V. and L.P.-O. led the manuscript writing with the input of all the authors; and L.P.-O. supervised the project. Correspondence to Ludmila Prokunina-Olsson (prokuninal@mail.nih.gov).

## Funding

## Competing interests

The authors declare no competing interests.

## Additional information

[1]Laboratory of Translational Genomics, DCEG, National Cancer Institute, Rockville, MD, USA. [2]Cancer Genomics Research Laboratory, Leidos Biomedical Research, Frederick National Laboratory for Cancer Research, Frederick, MD, USA. [3]Center for Alzheimer's and Related Dementias, National Institute of Aging and National Institute of Neurological Disorders and Stroke, Bethesda, MD, USA. [4]Cancer Data Science Laboratory, CCR, National Cancer Institute, Bethesda, MD, USA. [5]UC Santa Cruz Genomics Institute, Santa Cruz, CA, USA. [6]Genome Modification Core, Laboratory Animal Sciences Program, Leidos Biomedical Research, Frederick National Laboratory for Cancer Research, Frederick, MD, USA. [7]Chemical Biology Laboratory, CCR, National Cancer Institute, Frederick, MD, USA. [8]Integrative Tumor Epidemiology Branch, DCEG, National Cancer Institute, Rockville, MD, USA. [9]Laboratory of Genetic Susceptibility, DCEG, National Cancer Institute, Rockville, MD, USA. [10]Clinical Genetics Branch, DCEG, National Cancer Institute, Rockville, MD, USA. [11]Infections and Immunoepidemiology Branch, DCEG, National Cancer Institute, Rockville, MD, USA. ✉e-mail: prokuninal@mail.nih.gov

