## [Transparent Peer Review file · Nature Communications]

Genetic regulation of TERT splicing affects cancer risk by altering cellular longevity and replicative potential

Corresponding Author: Dr Ludmila Prokunina-Olsson

Version 0:

Reviewer comments:

Reviewer #1

(Remarks to the Author)

In this study, the authors described: 1.) the linkage between a SNP rs10069690 and a tandem repeat VNTR6-1; 2.) the potential functions of rs10069690 and VNTR6-1 in regulating the expression and splicing of TERT gene; 3.) the roles of VNTR6-1 in regulating cell proliferation and apoptosis; 4.) The linking of VNTR6-1 with SNPs that are associated some cancer types; 5.) the association of TERT isoforms and genetic variants with telomerase-related metrics; 6.) VNTR6-1 short and rs10069690-C only present in human genome. Deciphering the roles of SNPs and tandem repeats present in human genome is really important for us to decode the functionalities of regulatory elements across human genome, thus it is an area of intense investigation. However, in its current form the authors have not convinced this reviewer of the importance and functionality of rs10069690-T and VNTR6-1 long allele.

1. The author want to deliver to much information in the manuscript, which make the evidence is not solid enough to support the conclusions in some result sections:

1.) The author observed a high G content within VNTR6-1 repeat unit and concluded that “VNTR6-1 creates an expanded G quadruplex that modulates TERT splicing” without providing any other evidence to support the existence of G quadruplex. Considering the TERT-FL:TERT-beta ratio changed in WT and V6.1-KO cells after treated by G4-stabilizing ligands, it is hard for the reviewer to believe that VNTR6-1 regulates the splicing of TERT via creating an expanded G quadruplex;

2.) The author concluded that “rs10069690-T and VNTR6-1 long alleles affect TERT expression and splicing”, I didn't think VNTR6-1 long alleles and rs10069690-T could affect TERT expression (figure3d and 3e). The combination of rs10069690-T and VNTR6-1 might have some mild effect ($p=0.027$), but not strong effect as claimed by the authors; in lines 188-190, the authors stated “compared to Short-C haplotype, the TERT expression was decreased by the short-T and long-C”, I don't think the statement is appropriate since the results were not statistically significant.

3.) The authors concluded in line 226 that “These results suggest that the ratios of TERT-FL and TERT-beta affect both cellular longevity (by protecting cells from apoptosis)...”, however, the author did not provide any evidence to show the roles of two TERT isoforms in protecting the cells form apoptosis.

....., there were more and authors should be cautious to draw conclusions in their manuscript.

2. It is not clear how VNTR6-1 and rs10069690 reduce TERT expression by nonsense-mediated decay.

3. Why the cell growth increase the most at a 50:50% TERT-FL:TERT-beta ratio?

4. How does the length of VNTR6-1 related TERT isoforms?

(Remarks on code availability)

Reviewer #2

(Remarks to the Author)

This is a very interesting and experimentally extremely well conducted study using many state-of-the-Art methods uncovering and exploring a novel splice mechanisms of the TERT (telomerase) gene and its association with cancer, human evolution and longevity using a wealth of data bases. I sincerely congratulate the authors for such a wonderful study. However, there are a few minor issues that need to be addressed, mainly in some terminology and interpretation of results where some unsubstantiated statements and over-interpretations should be corrected. Please see details below:

1. Abstract, lines 49/50 and various statements throughout the ms (for example lines 123, 168, 291, 330 and others): A ratio always requires 2 players, for example, as correctly stated in line 119: TERT-FL to TERT beta. If you do not list both factors which would justify using "ratio", better use "percentage" or "fraction". Please amend.
2. Line 50 and other places: Please explain all abbreviations when they first appear in the text, here: TERT-FL, but also LD and PLCO in the main text and not just in the methods part or supplementary materials.
3. Line 51 and other places in the ms like for example, lines 196-209, line 218, 221, 222) : The term "cell growth" rather means the increase of cell mass (weight, volume etc.). You rather here refer to "cell proliferation" or "growth of cell population" with the former being the more commonly used term. Please correct.
4. Line 137: Where comes the "S" in HNSC for head and neck cancer from? Please double check whether the abbreviation is correct.
5. 148: Please explain "LD" (see above).
6. line 183: The statement requires a reference.
7. Lines 222/3: Only 3 different ratios were analysed, so the scaling of 2 from 3 seems a bit questionable. Perhaps consider rephrasing the sentence.
8. Line 225: Please describe "scattered localization of TERT-FL" in more detail. How do you explain that you do not find it in the nucleus where most cancer cells harbor it in order to maintain their telomeres. Please comments on its apparent absence in the nucleus.
9. Line 227: " RATIO (not plural!) of TERT FL TO TERT-beta" ...
10. Lines 240, 309 and others: In my view the term "infer" is not suitable here since it rather means "conclude". Perhaps you should rather use "employed" or something similar.
11. Line 256: Please explain the abbreviation "GTE_x" also in the main text (see comment above).
12. Line 277: The idiom "5 year age group" is not entirely correct grammar. Better say "5 year INTERVALL groups" or something similar.
13. Line 294: Have you explained the abbreviation PLCO before?
14. Lines 323/4: The statement is not clear and seems to lack in its generalising in direct evidence. How did you determine the longevity of non-proliferating (only neurons are postmitotic in brain tissue while glia cells are not) and what did you compare it with? Also, why does the higher percentage of the TERT beta splice form result in more somatic mutations and how is this related to a presumed higher cell lifespan (longevity is usually rather used for whole organisms). Please explain, specify and provide appropriate references. Do not overinterpret your own results. In my view this is just a wild speculation.
15. Line 329: The statement is in my view a direct contradiction. Please specify why you think (and what evidence you have for it) that an anti-apoptotic effect of TERT beta reduces proliferation? Why is this proliferation regenerative? Regeneration does not apply to cancer cells.
16. Lines 330/4: There is not such a thing like "replicative mutagenesis"! Perhaps better say "mutagenesis occurring due to replication errors" or something along these lines.
17. Line 331: Your assumption of a direct connection between a lower fraction of TERT-FL, presumed mutations/mutated cells (where do they come from in this context here?) and a smaller risk for immortalisation since the latter just refers to the canonical, telomere-dependent function of TERT which also requires the presence of TERC which is decreased/non-existent in neurons for example (Ishaq et al., 2016). In contrast, the non-canonical role of TERT in some post-mitotic cell types such as neurons is unrelated to increased mutation rates and immortalization, or at least has, to my knowledge, never been analysed yet) Please amend and correct your statements.
18. Lines 339/40: In your statement it is not clear how a critical telomere shortening should correlate to high proliferation rates since the former results in a cell cycle arrest and senescence or apoptosis. Please clarify and rephrase.
19. Line 345: Tert on its own has NO telomerase activity (TA), but for that it requires the assembly with TERC. Also, better specify that TA results in telomere maintenance while "cellular homeostasis" also exists in postmitotic cells and is far too general in the context of TA/telomeres. Please correct.
20. Lines 347-349: References 41 and 42 are both wrongly cited. NONE of these studies demonstrated anything on telomeres! For the correlation between telomere shortening and oxidative stress including telomeric DNA damage, please cite either von Zglinicki et al., 1995, Petersen et al., 1998 or von Zglinicki, 2002. References 41 and 42 both show that under increased oxidative stress TERT can be located into mitochondria whereby ox stress/ROS levels get decreased (which you correctly describe in lines 349/50 citing reference 41), however, ref. 41 as well as 42 also show that mitochondrial TERT decreases the sensitivity to apoptosis while reference 43 is not suitable since it is outdated and it has been later shown as incorrect by the same Santos group who later confirmed the results from 41 and 42 on decreased apoptosis. Thus, please correct these incorrect statements.
21. Regarding the role of TERT-splicing versions for adaptation to a microenvironment, it might be worth citing Radan et al., 2014 (doi: 10.1089/scd.2013.0373.) who showed that for stem cells under different oxygen conditions.
22. Figure legend for S14: I would discourage the use of the term "protein expression" since only genes are expressed. In order to avoid any confusion, I would rather suggest to use "protein levels"

(Remarks on code availability)

Version 1:

Reviewer comments:

Reviewer #1

(Remarks to the Author)

Thank you for addressing my comments. I don't have any further questions.

(Remarks on code availability)

Reviewer #2

(Remarks to the Author)

The authors addressed most of my comments and in general changed, added and improved a lot in their manuscript providing better details for the reader community for such a rather complex research topic.

The study adds a lot of significant details to this topic by carefully analysing various intra-intron TERT sequences and correlate them with different types of cancer.

I have only two minor additional comments:

1. In line 394 please change "telomerase" to "telomere"-independent. since the former might relate to telomerase activity but since often telomerase and TERT are used as synonyms this wording is rather ambiguous.

2. In line 400 you hypothesise that the beta TERT splice form might increase proliferation/replicative potential. in particular, under stress. While I agree that your data show a better survival of cells under stress due to decreased apoptosis sensitivity and higher proliferation rate, the beta form has no telomerase activity which is the most important prerequisite for ongoing proliferation potential in immortal and cancer cells. Thus, please clarify what exactly you mean with "replicative potential" and how this reconciles with the lack of enzymatic activity of the beta splice form. In my view, and you describe this correctly within the ms, There is an environment/stress-dependent balance between FL and beta form which seems to address the actual preferential needs of a cell or culture. This is a similar situation for even FL TERT when it is localized in the mitochondria during stressful conditions where it decreases ROS and apoptosis sensitivity, but cannot work on telomeres which it can do only when it is in the nucleus.

(Remarks on code availability)

I have checked the URL and it looks fine to me with readme files etc. However, I have no expertise in genome research. My field of expertise is telomerase and the current study adds a lot of significant details to this topic by carefully analysing various intra-intron TERT sequences and correlate them with different types of cancer.

Genetic regulation of TERT splicing contributes to reduced or elevated cancer risk by altering cellular longevity and replicative potential

Responses to Reviewers' Comments

Reviewer #1, expertise in long-read sequencing, Hi-ChIP, DNA methylation analysis, bioinformatics and systems biology (Remarks to the Author):

In this study, the authors described:

- 1.) the linkage between a SNP rs10069690 and a tandem repeat VNTR6-1;
- 2.) the potential functions of rs10069690 and VNTR6-1 in regulating the expression and splicing of TERT gene;
- 3.) the roles of VNTR6-1 in regulating cell proliferation and apoptosis;
- 5.) The linking of VNTR6-1 with SNPs that are associated some cancer types;
- 5.) the association of TERT isoforms and genetic variants with telomerase-related metrics;
- 6.) VNTR6-1 short and rs10069690-C only present in human genome.

Deciphering the roles of SNPs and tandem repeats present in human genome is really important for us to decode the functionalities of regulatory elements across human genome, thus it is an area of intense investigation. However, in its current form the authors have not convinced this reviewer of the importance and functionality of rs10069690-T and VNTR6-1 long allele.

1. The author want to deliver to much information in the manuscript, which make the evidence is not solid enough to support the conclusions in some result sections:

Response: We have carefully reviewed the manuscript and added further explanations and detailed results to strengthen our conclusions.

- a. The author observed a high G content within VNTR6-1 repeat unit and concluded that “VNTR6-1 creates an expanded G quadruplex that modulates TERT splicing” without providing any other evidence to support the existence of G quadruplex. Considering the TERT-FL:TERT-beta ratio changed in WT and V6.1-KO cells after treated by G4-stabilizing ligands, it is hard for the reviewer to believe that VNTR6-1 regulates the splicing of TERT via creating an expanded G quadruplex;

Response: The existence of G-quadruplexes (G4) is supported by several lines of evidence presented in supplementary materials. Specifically, we showed prediction with G4Hunter program (**Figure S11b**), results of G4-Seq (**Figure 2a**) and G4 ligand treatment (**Figure 2c-g, S8d-k**). We now provided this information in the main text as well, with additional explanations to ensure the evidence supporting our conclusions is clear.

- b. The author concluded that “rs10069690-T and VNTR6-1 long alleles affect TERT expression and splicing”, I didn't think VNTR6-1 long alleles and rs10069690-T could affect TERT

expression (figure 3d and 3e). The combination of rs10069690-T and VNTR6-1 might have some mild effect ($p=0.027$), but not strong effect as claimed by the authors; in lines 188-190, the authors stated “compared to Short-C haplotype, the *TERT* expression was decreased by the short-T and long-C”, I don’t think the statement is appropriate since the results were not statistically significant.

Response: Compared to the Short-C haplotype, the *TERT* expression was decreased by the Short-T ($\beta=-12.2$ TPMs, $p=0.10$) and Long-C ($\beta=-15.92$ TPMs, $p=0.36$) haplotypes, with a greater decrease occurring when both the VNTR6-1-Long and rs10069690-T alleles were included in the same haplotype (Long-T, $\beta=-24.18$ TPMs, $p=0.027$). As we mentioned, the expression analysis of *TERT* is extremely difficult due to its generally very low expression. We were fortunate to identify and get access to the Burkitt lymphoma dataset as a unique, although limited, source of samples with high *TERT* expression and without the somatic promoter mutations that occur in most other cancer types. Our haplotype-based analysis that has never been done for *TERT* expression before uncovered associations that are both statistically significant and biologically relevant,

- c. The authors concluded in line 226 that “These results suggest that the ratios of *TERT*-FL and *TERT*-beta affect both cellular longevity (by protecting cells from apoptosis)....”, however, the author did not provide any evidence to show the roles of two *TERT* isoforms in protecting the cells from apoptosis.
....., there were more and authors should be cautious to draw conclusions in their manuscript.

Response: We observed shifts in *TERT* isoform ratio (*TERT*-FL:*TERT*- β) toward higher *TERT*-FL fraction when VNTR6-1 was eliminated by CRISPR-Cas9 editing initially in three independent clones of UMUC3 cells and now in two independent clones of A549 cells as well (Figures 2b, e; current Figures S7c, e, f and S8b-c). Follow-up functional analyses (Figure 4d-e and now also in an additional Figure S18b-c) further demonstrated that this knockout was associated with increased apoptosis when cells were cultured with either cisplatin or in full serum media, in all 5 independent clones of 2 cell lines tested. RNA-seq analysis also showed that genes in the apoptosis resistance pathway were predominantly downregulated after VNTR6-1 knockout. These results suggest that the presence of VNTR6-1 in WT cells (which has a higher fraction of *TERT*- β Figures 2b, e; current Figures S7c, e, f and S8b-c), helps protect cells against apoptosis. Additionally, the coexpression of both isoforms at the *TERT*-FL:*TERT*- β ratio of 20:80% or 80:20% (approximately modeling the WT vs. V6.1-KO cell lines), resulted in slower cell growth (as shown by statistical significance in Table S12), further supporting the observed increase in apoptosis. Our microscopy results showed stronger mitochondrial localization for *TERT*- β than for *TERT*-FL, placing *TERT*- β in the subcellular location where apoptosis is regulated.

Collectively, these findings based on several independent methods provide compelling evidence to support a critical role for *TERT*- β rather than *TERT*-FL in apoptosis protection and

proliferation. However, we remain cautious in our interpretations, aiming to reconcile both the genetic and functional observations with new insights.

2. It is not clear how VNTR6-1 and rs10069690 reduce TERT expression by nonsense-mediated decay.

Response: The nonsense-mediated decay (NMD) affects prematurely terminated transcripts. NMD is likely to be partial and regulated by additional factors, but this is outside the scope of the paper. We provide this information: “Transcripts truncated by premature termination codons (**Figure S12**), including *INS* (truncated within exon 5), *INS1b* (intron 4), and *TERT-β* or *TERT-αβ* (exon 10), are likely to be eliminated by nonsense-mediated decay (NMD), reducing total *TERT* expression”.

3. Why the cell growth increase the most at a 50:50% TERT-FL:TERT-beta ratio?

Response: A 50:50% TERT-FL:TERT-β ratio may provide a better balance between cell growth and apoptosis protection but detailed investigations into all the factors affecting cell growth are beyond the scope of this study. In our updated manuscript, we have removed the 50:50% condition as optimizing growth in an overexpression model is not relevant to the point of this analysis and adds confusion. We now focus on more extreme ratios, 20: 80% and 80:20% of TERT-FL:TERT-β plasmids.

We provide the following text: “We monitored proliferation in a bladder cancer cell line with low *TERT* expression (5637 cells, DepMap *TERT* TPM=1.23) after transient transfection with the *TERT-FL* or *TERT-β* plasmid (**Figure S19a, b**). Compared to the GFP control, overexpression of either isoform increased proliferation, with a weaker effect for *TERT-FL* compared to *TERT-β* (**Figure S19c, Table S12**). Co-transfection at 20:80% and 80:20% *TERT-FL:TERT-β* plasmid ratios (modeling WT and V6.1-KO cells, respectively) also increased proliferation compared to control. However, cells transfected with more *TERT-FL* (80:20% ratio) grew slower than those transfected with more *TERT-β* (20:80% ratio, **Figure S19d, Table S12**), potentially due to reduced levels of anti-apoptotic TERT-β.”

4. How does the length of VNTR6-1 related TERT isoforms?

Response: Longer VNTR6-1 has more G4 copies (35 vs. 113 for the shortest vs. longest repeats). By creating extended regions of stable secondary structure and likely DNA:RNA hybrids, more G4 copies might interfere with the splicing of TERT exons 7-8, causing exon skipping and generation of the *TERT-β* isoform. The deletion of VNTR6-1 increased the inclusion of exons 7 and 8 (**Figure 2b, e, S7c-f, S8b, c**), both in UMUC3 and A549 cell lines (new data), suggesting that VNTR6-1 acts as a splicing switch between the *TERT-FL* (higher

ratio in V6.1-KO) and *TERT-β* (higher ratio in WT cells).

Reviewer #2, expertise in TERT genetics and telomerase biology (Remarks to the Author):

This is a very interesting and experimentally extremely well conducted study using many state-of-the-Art methods uncovering and exploring a novel splice mechanisms of the TERT (telomerase) gene and its association with cancer, human evolution and longevity using a wealth of data bases. I sincerely congratulate the authors for such a wonderful study. However, there are a few minor issues that need to be addressed, mainly in some terminology and interpretation of results where some unsubstantiated statements and over-interpretations should be corrected. Please see details below:

1. Abstract, lines 49/50 and various statements throughout the ms (for example lines 123, 168, 291, 330 and others): A ratio always requires 2 players, for example, as correctly stated in line 119: TERT-FL to TERT beta. If you do not list both factors which would justify using "ratio", better use "percentage" or "fraction". Please amend.

Response: We thank the reviewer for the thorough and supportive comments and made the suggested revisions. We have replaced instances of “ratio” with “fraction” where only one factor was referenced, ensuring consistency and clarity, as advised.

2. Line 50 and other places: Please explain all abbreviations when they first appear in the text, here: TERT-FL, but also LD and PLCO in the main text and not just in the methods part or supplementary materials.

Response: The text was edited and checked.

Current lines: 149 for TERT-FL (*TERT*-full-length), 129-130 for LD (linkage disequilibrium), and 312 for PLCO (Prostate, Lung, Colorectal and Ovarian)

3. Line 51 and other places in the ms like for example, lines 196-209, line 218, 221, 222) : The term "cell growth" rather means the increase of cell mass (weight, volume etc.). You rather here refer to "cell proliferation" or "growth of cell population" with the former being the more commonly used term. Please correct.

Response: Thank you for this clarification. We have edited the text to replace “cell growth” with “cell proliferation” where appropriate, aligning terminology with standard usage. For clarity, we now use “proliferation” in our overall observations and conclusions based on xCELLigence/Lionheart data, which provide live cell counts. We specify in the text that the “cell index” measured by xCELLigence is a measure of cell proliferation.

We have also added the term “cell division” to specifically refer to CFSE assay results and conclusions, as this assay measures dye depletion that occurs with every cell division. We use “cell division” to differentiate from “proliferation,” which reflects the balance between cell

division and cell death.

4. Line 137: Where comes the "S" in HNSC for head and neck cancer from? Please double check whether the abbreviation is correct.

Response: The text was corrected to “head and neck squamous carcinoma, HNSC”, which is an official TCGA abbreviation for this cancer.

5. 148: Please explain "LD" (see above).

Response: Checked and it was mentioned accordingly, lines 129-130.

6. line 183: The statement requires a reference.

Response: Reference is added (PMID: 23610451).

7. Lines 222/3: Only 3 different ratios were analysed, so the scaling of 2 from 3 seems a bit questionable. Perhaps consider rephrasing the sentence.

Response: Thank you for the suggestions. The text is edited, now mentioning only 20:80% and 80:20% ratios, lines 287-289.

8. Line 225: Please describe "scattered localization of TERT-FL" in more detail. How do you explain that you do not find it in the nucleus where most cancer cells harbor it in order to maintain their telomeres. Please comments on its apparent absence in the nucleus.

Response: The cell image presented in the previous version of the paper happened to have a lower nuclear expression of TERT (hardly visible in the image), but we chose it for its mitochondrial reference marker (TOM20) and positive signals from our expression constructs detected by well-validated antibodies for FLAG and HA protein tags (epitopes added to TERT- β and TERT-FL constructs, respectively). It is possible that the transfection and overall protein expression were low for imaging, but we acknowledge that the lack of nuclear expression did not line up with what is expected for TERT-FL and the lack of mitochondrial localization for TERT-FL could not be concluded without the expected TERT-FL expression. In repeat experiments, we focused on maximizing the transfection efficiency and double-checking constructs, controls, laser settings, alignment, etc. before and during imaging.

We noted that when the transfection efficiency was high, with most cells expressing both tagged proteins higher than the background, cells with strong nuclear expression of TERT-FL-HA were quite rare. When we zoomed out to scan the slides, we identified cells with high expression of TERT-FL-HA and TERT- β -FLAG, in the nucleus (see below, **provided to reviewers only**). However, these were typically single cells within a cluster of other transfected cells, where most

showed no strong nuclear TERT expression. This could be due to the cells being stressed during overexpression, leading to TERT being more readily exported from the nucleus, as reported in the literature (PMID: 12808100). Exploring this interesting observation is beyond the scope of this already busy manuscript. We repeated the overexpression transfections and staining multiple times, with the same observations each time. We validated all our constructs by Sanger sequencing. As we could not find a specific and reliable antibody for the detection of endogenous TERT, we have to use the overexpression system. In our updated manuscript, we have included images of a cell with the expected nuclear TERT-FL-HA expression from repeated experiments in a new main figure (**Figure 5a**) and a supplementary figure (**Figure S20**).

Provided to reviewers only:

Provided to reviewers only: Confocal images of clusters of *TERT-FL/TERT-β* transfected cells, showing that in efficiently transfected cells (*TERT-FL-HA* and *TERT-β-FLAG* signals present in nearly all cells in view), strong nuclear expression of *TERT-FL* is rare. Examples of three different zoomed-out fields (40x magnification) with multiple cells per field show this is not an isolated event.

New figures showing both TERT-FL-HA with nuclear expression and co-localization of TERT- β -FLAG with mitochondria that is stronger than that for TERT-FL-HA:

Figure 5. The functional differences between the TERT-FL and TERT- β isoforms.

a, Structured illumination microscopy images of A549 cells cotransfected with TERT-FL and TERT- β expression constructs at a 50:50% ratio. For individual channels, staining is shown as black/white images for better contrast. On tri-color merged panels, green – FLAG (TERT- β) or HA (TERT-FL), blue – DAPI (nuclei). On the quad-color merged panel, purple - HA (TERT-FL), green - FLAG (TERT- β), red - TOM20 (mitochondria), blue - DAPI (nuclei). The yellow inset in the TERT- β -FLAG panel is shown at a higher magnification to demonstrate colocalization with mitochondria (yellow staining).

Figure S20. Cellular localization of the TERT-FL and TERT- β protein isoforms.

Structured illumination microscopy (SIM) images of the TERT-FL and TERT- β protein isoforms transiently overexpressed in the A549 lung cancer cell line. Cells were co-transfected with TERT-FL-HA and TERT- β -FLAG expression constructs at a 50:50% ratio and stained with corresponding antibodies. For individual channels, staining is shown as black/white images for better contrast. On tri-color merged panels, green – FLAG (TERT- β) or HA (TERT-FL), blue – DAPI (nuclei). On the quad-color merged panel, purple – HA (TERT-FL), green – FLAG (TERT- β), red – TOM20 (mitochondria), blue – DAPI (nuclei). **a**, Zoomed-out view of transfected cells stained with anti-FLAG and anti-TOM20 primary antibodies, as well as respective secondary antibodies AlexaFluor647-/AlexaFluor488- (pseudocolored green) and AlexaFluor555- (pseudocolored red). The orange color indicates the mitochondrial colocalization of TOM20 with TERT- β , not seen for TERT-FL. The region in the cyan box is zoomed-in in **Figure 5a**. **b**, View of negative controls: similarly transfected cells but omitting primary antibodies and stained only

with secondary antibodies, confirming that the detected signals are not from autofluorescence. Images were captured with a 63x/1.4 NA objective. Scale bar = 2 μ m.

9. Line 227: " RATIO (not plural!) of TERT FL TO TERT-beta"...

Response: Edited in multiple places.

10. Lines 240, 309 and others: In my view the term "infer" is not suitable here since it rather means "conclude". Perhaps you should rather use "employed" or something similar.

Response: We would like to keep the term “infer” to differentiate it from “impute”. Both terms refer to statistical approaches to derive genotype information based on other markers.

11. Line 256: Please explain the abbreviation "GTEx" also in the main text (see comment above).

Response: We made sure the term is explained at the first mention, line 183

12. Line 277: The idiom "5 year age group" is not entirely correct grammar. Better say "5 year INTERVALL groups" or something similar.

Response: Edited, line 346

13. Line 294: Have you explained the abbreviation PLCO before?

Response: We made sure the term is explained at the first mention, line 311

14. Lines 323/4: The statement is not clear and seems to lack in its generalising in direct evidence. How did you determine the longevity of non-proliferating (only neurons are postmitotic in brain tissue while glia cells are not) and what did you compare it with? Also, why does the higher percentage of the TERT beta splice form result in more somatic mutations and how is this related to a presumed higher cell lifespan (longevity is usually rather used for whole organisms). Please explain, specify and provide appropriate references. Do not overinterpret your own results. In my view this is just a wild speculation.

Response: We appreciate the comment. We acknowledge that our proposed model is based on genetic associations with cancer risk, telomere length and functional data, and we cannot possibly test and verify all the components of this model in a single paper. We also took the liberty of using the term “cellular longevity,” referring to the lifespan of individual cells, as was also used in PMID: 12232501.

We agree with the comment and have removed the sentence with the two mentioned statements, “longevity of such non-proliferating cells increases...” and “a higher ratio of the *TERT- β* isoform can result in the accumulation of somatic mutations.” Glioma is one of the cancers with the

strongest associations for the genetic variants tested, and its likely origin is glial cells, that are not post-mitotic but still with low replicative potential.

15. Line 329: The statement is in my view a direct contradiction. Please specify why you think (and what evidence you have for it) that an anti-apoptotic effect of TERT beta reduces proliferation? Why is this proliferation regenerative? Regeneration does not apply to cancer cells.

Response to comments 14 and 15: We appreciate the thoughtful comments and the opportunity to clarify our points. We define regenerative proliferation as proliferation induced by tissue damage, in contrast to homeostatic proliferation, which maintains normal tissue renewal. Since we explore mechanisms of cancer susceptibility, that is, transition from normal to cancer state, the mechanisms we discuss predominantly affect normal tissues.

In this context, homeostasis is central to the tissue, as stem cells and TERT^{high} cells undergo continuous turnover to maintain a stable internal environment. This turnover rate varies significantly across tissues – occurring more frequently in tissues such as the gastrointestinal tract, and much more slowly in others, such as the brain or bladder. In response to endogenous or exogenous exposures and stressors causing tissue injury, this turnover process accelerates to facilitate repair. However, each cycle of cell turnover carries the potential for mutations due to replication errors.

Tissue repair is tightly regulated by homeostatic mechanisms that prevent excessive and unnecessary cellular turnover, and each organ responds to environmental stressors according to its unique physiological and anatomical characteristics. In our association analysis, cancers of the bladder and brain (both low-proliferative tissues in normal conditions) showed the strongest effects in opposite directions, helping to illustrate how cancer risk can be modified. While the bladder is more vulnerable to the direct effects of environmental contaminants through urine, the brain limits direct exposure to such insults through the protective blood-brain barrier. In this context, the cellular features affected by TERT- β and TERT-FL may play a crucial role.

Specifically, the demonstrated anti-apoptotic effect of TERT- β might play different roles in these tissues. In highly exposed tissues with low homeostatic proliferation but high regenerative proliferation in response to tissue damage, such as the bladder, the antiapoptotic TERT- β could reduce cell death. The decreased need for regeneration could limit the accumulation of mutations from replication, on top of those caused by endogenous or exogenous exposure. This could potentially reduce cancer risk. In contrast, in tissues with limited exposure and low regenerative proliferation, such as glial cells in the brain, where the blood-brain barrier protects against immediate damage, the same anti-apoptotic effect of TERT- β could be harmful. By extending the lifespan of these cells, TERT- β might allow the accumulation of somatic mutations over time, increasing the likelihood of cellular transformation and potentially increasing the risk of cancer.

We also acknowledge the role of reciprocally altered TER-FL fraction:

While the anti-apoptotic function of TERT- β is important, the reciprocal decrease in the TERT-FL fraction, also associated with the VNTR6-1-Long and rs10069690-T alleles, might prevent immortalization of cells with acquired somatic mutations or protect against telomere shortening, especially under oxidative stress^{42,48} and facilitate the DNA damage response⁴⁹.

16. Lines 330/4: There is not such a thing like "replicative mutagenesis"! Perhaps better say "mutagenesis occurring due to replication errors" or something along these lines.

Response: We appreciate the suggestion and have revised the statement to read as “mutagenesis from replication errors”, line 414

17. Line 331: Your assumption of a direct connection between a lower fraction of TERT-FL, presumed mutations/mutated cells (where do they come from in this context here?) and a smaller risk for immortalisation since the latter just refers to the canonical, telomere-dependent function of TERT which also requires the presence of TERC which is decreased/non-existent in neurons for example (Ishaq et al., 2016). In contrast, the non-canonical role of TERT in some post-mitotic cell types such as neurons is unrelated to increased mutation rates and immortalization, or at least has, to my knowledge, never been analysed yet) Please amend and correct your statements.

Response: Thank you for your comment. In the updated version, we have clarified that the canonical role of TERT is mediated by the TERT-TERC complex in telomerase activity. Additionally, we have reworded the section to highlight the non-canonical roles of TERT, specifically focusing on the TERT- β isoform's potential influence on cancer risk or protection through a telomerase-independent anti-apoptotic function, likely linked to its mitochondrial localization.

Since the TERT-FL and TERT- β fractions are reciprocal, we added a statement that not only an increase in one fraction but also a decrease in the other fraction is important. For example, the increase in TERT-FL in bladder tissue might increase the immortalization of mutated cells, while in the glial cells, the deficiency of TERT-FL might be detrimental.

18. Lines 339/40: In your statement it is not clear how a critical telomere shortening should correlate to high proliferation rates since the former results in a cell cycle arrest and senescence or apoptosis. Please clarify and rephrase.

Response: Thank you for your feedback. We have addressed this point in the revised text, specifically in lines 429-430: “High proliferation rates in stem cells of these tissues, combined with cell death induced by critical telomere shortening in differentiated cells, prevent cells from reaching a malignant state and thus act as a tumor-suppressive mechanism⁴⁹”

19. Line 345: Tert on its own has NO telomerase activity (TA), but for that it requires the assembly with TERC. Also, better specify that TA results in telomere maintenance while "cellular

homeostasis" also exists in postmitotic cells and is far too general in the context of TA/telomeres. Please correct.

Response: Thank you for your feedback. In the revised text, we have clarified that telomerase activity relies on the TERT-TERC complex, in lines 392-394:

In addition to its canonical role in telomerase activity that is mediated by the TERT-TERC complex and supports telomere maintenance, TERT also has important non-canonical telomerase-independent roles in supporting cellular homeostasis.

20. Lines 347-349: References 41 and 42 are both wrongly cited. NONE of these studies demonstrated anything on telomeres! For the correlation between telomere shortening and oxidative stress including telomeric DNA damage, please cite either von Zglinicki et al., 1995, Petersen et al., 1998 or von Zglinicki, 2002. References 41 and 42 both show that under increased oxidative stress TERT can be located into mitochondria whereby ox stress/ROS levels get decreased (which you correctly describe in lines 349/50 citing reference 41), however, ref. 41 as well as 42 also show that mitochondrial TERT decreases the sensitivity to apoptosis while reference 43 is not suitable since it is outdated and it has been later shown as incorrect by the same Santos group who later confirmed the results from 41 and 42 on decreased apoptosis. Thus, please correct these incorrect statements.

Response: Thank you for the suggestion. The references in the discussion were updated using the von Zglinicki et al. 1995 paper.

21. Regarding the role of TERT-splicing versions for adaptation to a microenvironment, it might be worth citing Radan et al., 2014 (doi: 10.1089/scd.2013.0373.) who showed that for stem cells under different oxygen conditions.

Response: Thank you for the suggestion. It makes some interesting points regarding a functional role for TERT-beta in stem cell maintenance, which may be useful to our discussion about tissues with low replicative potential. However, due to word limitations, we decided not to expand the discussion into this topic.

22. Figure legend for S14: I would discourage the use of the term "protein expression" since only genes are expressed. In order to avoid any confusion, I would rather suggest to use "protein levels".

Response: Thank you for the suggestion. Edited to "protein levels", rearranged to Figure S19.

Additional results included in the updated version

1. Additional analyses to test VNTR6-1 against all cancer-related GWAS signals in the region (Table S1)
2. Additional details on the machine learning strategy used to select the two proxy SNPs that capture VNTR6-1 (Figure S3).

Figure S3. Random forest machine learning analysis of SNP-based prediction of VNTR6-1 groups in 3,201 individuals from the 1000G.

a, The top 10% of SNPs within the 400 kb genomic region (GRCh38 chr5:1,100,000-1,500,000) were selected based on Mean Decrease in Gini values, indicating their high discriminative power to predict VNTR6-1 Short/Short and Long/any genotype groups, which were established for each sample based on short-read WGS profiles. Dot sizes represent a Mean Decrease in Accuracy, with larger circles corresponding to higher model accuracy and higher values corresponding to substantial accuracy loss if the feature is removed. SNPs with both a high Mean Decrease in Gini and a high Mean Decrease in Accuracy are the most informative for distinguishing the VNTR6-1 groups. **b**, A representative decision tree from the ensemble of 500 trees in the random forest model illustrates the classification of VNTR6-1 groups. Each node in the tree represents a decision point based on the values of the predictor variables, leading to sample assignment either to VNTR6-1 Short/Short or Long/any groups.

3. Results of VNTR6-1 elimination by CRISPR-Cas editing in an additional cell line, A549 (**Figure S8**)

Figure S8. Splicing effects of *TERT*-VNTR6-1 knockout in the lung cancer cell line A549.

a, Agarose gels of PCR products amplified from genomic DNA of A549; *HBB* amplicon - normalization control. **b**, Agarose gels of RT-PCR products amplified from cDNA of corresponding samples; gDNA-genomic DNA, negative control; *HPRT1* - normalization control. **c**, Densitometry results of the PCR amplicons in plot **b**. Experiments in A549 cells comparing

TERT splicing and isoform-specific expression after 72 hours of treatment with G4 stabilizing ligands, normalized to *HPRT1* as an endogenous control in the WT (d, e) and VNTR6-1-KO (f, g) cell lines. d, f, A representative agarose gel of SYBR-Green RT-qPCR products detecting several isoforms with primers located in exons 6 and 9. The extra PCR band, marked by an arrow in panels c and d, is further explored in Figure S12. e, g, Densitometry analysis of the corresponding agarose gels evaluating the percentage of *TERT-FL* (%) relative to the total PCR products. h, j, Isoform-specific TaqMan RT-qPCR analysis of *TERT-FL* and *TERT-β* following treatment with G4-stabilizing ligands CX-5461 and PhenDC3 for 72 hours. i, k, Total *TERT* expression measured by TaqMan RT-qPCR assay for exon 3-4. All analyses are based on one of three representative experiments. Comparisons were made against the vehicle control (DMSO). All analyses are based on three experiments, with one representative gel shown. Comparisons were made against the vehicle control (DMSO). Statistical significance is indicated as follows: * $p < 0.05$, ** $p < 0.01$, *** $p < 0.001$, **** $p < 0.0001$, Student's T-test.

4. Effects of *TERT* VNTR6-1 knockout on proliferation and apoptosis in an additional cell line, A549 (Figure S18)

Figure S18. Effects of *TERT* VNTR6-1 knockout on proliferation and apoptosis in A549 cells.

a, Analysis of real-time increase in cell counts (cell index) measured with xCELLigence over 275 hours in A549 cells. The WT cells and V6.1-KO clone #2 (starred samples in **Table S9**) were cultured in CS medium for 48 hours, followed by the switch to fresh CS or full medium for 10 more days. Proliferation rates in response to culturing conditions were significantly decreased in VNTR6.1-KO compared to WT cells. Statistical significance and β -values for differences in the cell index during the visually determined growth phase (gray highlighting between 72 and 250 hours) were calculated using linear mixed-effects interaction models based on five replicates per sample. The reference sample is labeled with a dotted circle; β_{int} represents the change in growth rates between experimental groups. Quantification of apoptosis in WT and VNTR-6.1-KO A549 cells cultured for 48 hours in **b**, 10 μM cisplatin in CS medium or **c**, in full medium, followed by Annexin V/PI staining to determine the percentage of apoptotic cells in three replicates, normalized to CS media values and WT groups, * $p < 0.05$, ** $p < 0.01$, *** $p < 0.001$, Student's T-test. All data shown represent one of three independent experiments.

REVIEWERS' COMMENTS

Reviewer #1 (Remarks to the Author):

Thank you for addressing my comments. I don't have any further questions.

Response: We thank the reviewer for their contribution that improved our paper.

Reviewer #2 (Remarks to the Author):

The authors addressed most of my comments and in general changed, added and improved a lot in their manuscript providing better details for the reader community for such a rather complex research topic.

The study adds a lot of significant details to this topic by carefully analysing various intra-intron TERT sequences and correlate them with different types of cancer.

Response: We thank the reviewer for their thoughtful comments and suggestions that significantly improved our paper.

I have only two minor additional comments:

1. In line 394 please change "telomerase" to "telomere"-independent. since the former might relate to telomerase activity but since often telomerase and TERT are used as synonyms this wording is rather ambiguous.

Response: Thank you, telomerase-independent is changed to telomere-independent.

2. In line 400 you hypothesise that the beta TERT splice form might increase proliferation/replicative potential. in particular, under stress. While I agree that your data show a better survival of cells under stress due to decreased apoptosis sensitivity and higher proliferation rate, the beta form has no telomerase activity which is the most important prerequisite for ongoing proliferation potential in immortal and cancer cells. Thus, please clarify what exactly you mean with "replicative potential" and how this reconciles with the lack of enzymatic activity of the beta splice form. In my view, and you describe this correctly within the ms, There is an environment/stress-dependent balance between FL and beta form which seems to address the actual preferential needs of a cell or culture. This is a similar situation for even FL TERT when it is localized in the mitochondria during stressful conditions where it decreases ROS and apoptosis sensitivity, but cannot work on telomeres which it can do only when it is in the nucleus.

Response: We greatly appreciate the stimulating scientific insights of the reviewer that helped us in interpreting and presenting our results.

We define “replicative potential” as increased proliferation ($\ln 400$). In turn, because proliferation reflects the balance between cell division and apoptosis, we analyzed both processes ($\ln 269$). Since TERT-beta protects cells from apoptosis, it indirectly contributes to replicative potential, presumably through mechanisms unrelated to telomerase enzymatic activity.

We added the word “manifesting” to the sentence:

The increase in the fraction of the anti-apoptotic *TERT- β* isoform extends cellular longevity (lifespan of individual cells)⁴³, **manifesting as** increased proliferation (replicative potential), especially in response to cellular stress and other stimuli.

Reviewer #2 (Remarks on code availability):

I have checked the URL and it looks fine to me with readme files etc. However, I have no expertise in genome research. My field of expertise is telomerase and the current study adds a lot of significant details to this topic by carefully analysing various intra-intron TERT sequences and correlate them with different types of cancer.